# Boosting Alignment for Post-Unlearning Text-to-Image Generative Models

**Myeongseob Ko**[*]
Virginia Tech
`myeongseob@vt.edu`

**Henry Li**[*]
Yale University
`henry.li@yale.edu`

**Zhun Wang**
University of California, Berkeley
`zhun.wang@berkeley.edu`

**Jonathan Patsenker**
Yale University
`jonathan.patsenker@yale.edu`

**Jiachen T. Wang**
Princeton University
`tianhaowang@princeton.edu`

**Qinbin Li**
University of California, Berkeley
`liqinbin1998@gmail.com`

**Ming Jin**
Virginia Tech
`jinming@vt.edu`

**Dawn Song**
University of California, Berkeley
`dawnsong@berkeley.edu`

**Ruoxi Jia**
Virginia Tech
`ruoxijia@vt.edu`

## Abstract

Large-scale generative models have shown impressive image-generation capabilities, propelled by massive data. However, this often inadvertently leads to the generation of harmful or inappropriate content and raises copyright concerns. Driven by these concerns, machine unlearning has become crucial to effectively purge undesirable knowledge from models. While existing literature has studied various unlearning techniques, these often suffer from either poor unlearning quality or degradation in text-image alignment after unlearning, due to the competitive nature of these objectives. To address these challenges, we propose a framework that seeks an optimal model update at each unlearning iteration, ensuring monotonic improvement on both objectives. We further derive the characterization of such an update. In addition, we design procedures to strategically diversify the unlearning and remaining datasets to boost performance improvement. Our evaluation demonstrates that our method effectively removes target classes from recent diffusion-based generative models and concepts from stable diffusion models while maintaining close alignment with the models' original trained states, thus outperforming state-of-the-art baselines. Our code will be made available at `https://github.com/reds-lab/Restricted_gradient_diversity_unlearning.git`.

## 1 Introduction

Large-scale text-to-image generative models have recently gained considerable attention for their impressive image-generation capabilities. Despite being at the height of their popularity, these models, trained on vast amounts of public data, inevitably face concerns related to harmful content generation [Heng and Soh, 2024] and copyright infringement [Zhang et al., 2023b]. Although

---

[*]Equal contributions

38th Conference on Neural Information Processing Systems (NeurIPS 2024).

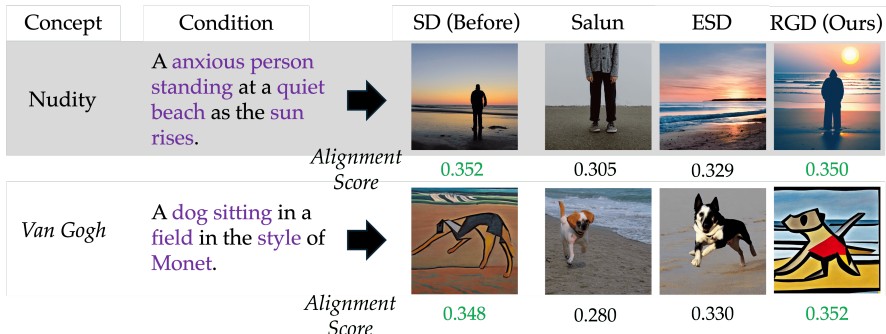

Figure 1: Generated images using `SalUn` [Fan et al., 2023], `ESD` [Gandikota et al., 2023], and Ours after unlearning given the condition. Each row indicates different unlearning tasks: nudity removal, and *Van Gogh* style removal. Generated images from our approach and `SD` [Rombach et al., 2022] are well-aligned with the prompt, whereas `SalUn` and `ESD` fail to generate semantically correct images given the condition. On average, across 100 different prompts, `SalUn` shows the lowest clip alignment scores (0.305 for nudity removal and 0.280 for *Van Gogh* style removal), followed by `ESD` (0.329 and 0.330, respectively). Our approach achieves scores of 0.350 and 0.352 for these tasks, closely matching the original `SD` scores of 0.352 and 0.348.

exact machine unlearning—retraining the model by excluding target data—is a direct solution, its computational challenge has driven continued research on approximate machine unlearning.

To address this challenge, recent studies [Fan et al., 2023, Gandikota et al., 2023, Heng and Soh, 2024], have introduced approximate unlearning techniques aimed at boosting efficiency while preserving effectiveness. These approaches have successfully demonstrated the ability to remove target concepts while maintaining the model's general image generation capabilities, with generation quality assessed using the Fréchet Inception Distance. However, these studies generally overlook the impact of unlearning on image-text alignment, which pertains to the semantic accuracy between generated images and their text descriptions [Lee et al., 2024]. While pretrained generative models generally demonstrate high alignment scores, our study reveals a critical gap: state-of-the-art unlearning techniques fall short in achieving comparable text-image alignment scores after unlearning, as illustrated in Figure 1. This could lead to potentially problematic behaviors in real-world deployments, necessitating further investigation.

We attribute the failure of existing techniques to maintain text-image alignment to two primary factors. Firstly, the unlearning objective often conflicts with the goal of maintaining low loss on the retained data, illustrating the competitive nature of these two objectives. Traditionally, approaches to optimizing these objectives have simply aggregated the gradients from both; however, this method of updating the model typically advances one objective at the expense of the other. Hence, while these approaches may successfully remove target concepts, they often compromise text-image alignment for retained concepts in the process. Secondly, current methods employ a simplistic approach to constructing a dataset for loss minimization on retained concepts. For example, in Fan et al. [2023], this dataset is composed of images generated from a single prompt associated with the concept to be removed. This lack of diversity in the dataset might lead to overfitting, which in turn hampers the text-image alignment.

To address these issues, we propose a principled framework designed to optimally balance the objectives of unlearning the target data and maintaining performance on the remaining data at each update iteration. Specifically, we introduce the concept of the *restricted gradient*, which allows for the optimization of both objectives while ensuring monotonic improvement in each objective. Furthermore, we have developed a deliberate procedure to enhance data diversity, preventing the model from overfitting to the limited samples in the remaining dataset. To the best of our knowledge, the strategic design of the forgetting target and remaining sets has not been extensively explored in the existing machine unlearning literature. In our evaluation, we demonstrate the improvement in both forgetting quality and alignment on the remaining data, compared to baselines. For example, our

evaluation in nudity removal demonstrates that our method effectively reduces the number of detected body parts to zero, compared to 598 with the baseline stable diffusion (SD) [Rombach et al., 2022], 48 with erased stable diffusion (ESD-u), and 3 with saliency map-based unlearning (SalUn) [Fan et al., 2023]. Particularly, while achieving this effective erasing performance, our method reduces the alignment gap to SD by 11x compared to ESD-u and by 20x compared to SalUn on the retained test set.

## 2 Related Work

### 2.1 Machine Unlearning

Machine unlearning has primarily been propelled by the "Right to be Forgotten" (RTBF), which upholds the right of users to request the deletion of their data. Given that large-scale models are often trained on web-scraped public data, this becomes a critical consideration for model developers to avoid the need for retraining models with each individual request. In addition to RTBF, recent concerns related to copyrights and harmful content generation further underscore the necessity and importance of in-depth research in machine unlearning. The principal challenge in this field lies in effectively erasing the target concept from pre-trained models while maintaining performance on other data. Recent studies have explored various approaches to unlearning, including the exact unlearning method [Bourtoule et al., 2021] and approximate methods such as using negative gradients, fine-tuning without the forget data, editing the entire parameter space of the model [Golatkar et al., 2020]. To encourage the targeted impact in the parameter space, [Golatkar et al., 2020, Foster et al., 2024] proposed leveraging the Fisher information matrix, and [Fan et al., 2023] leveraged a gradient-based weight saliency map to identify crucial neurons, thus minimizing the impact on remaining neurons. Furthermore, data-influence-based debiasing and unlearning have also been proposed [Chen et al., 2024, Bae et al., 2023]. Another line of work leverages mathematical tools in differential privacy [Guo et al., 2019, Chien et al., 2024] to ensure that the model's behavior remains indistinguishable between the retrained and unlearned models.

### 2.2 Machine Unlearning in Diffusion Models

Recent advancements in text-conditioned generative models [Ho and Salimans, 2022, Rombach et al., 2022], trained on extensive web-scraped datasets like LAION-5B [Schuhmann et al., 2022], have raised significant concerns about the generation of harmful content and copyright violations. A series of studies have addressed the challenge of machine unlearning in diffusion models [Heng and Soh, 2024, Gandikota et al., 2023, Zhang et al., 2023a, Fan et al., 2023]. One approach [Heng and Soh, 2024] interprets machine unlearning as a continual learning problem, showing effective removal results in classification tasks by employing Bayesian approaches to continual learning [Kirkpatrick et al., 2017], which enhance unlearning quality while maintaining model performance using generative reply [Shin et al., 2017]. However, this approach falls short in removing concepts such as nudity compared to other methods [Gandikota et al., 2023]. Another proposed method [Gandikota et al., 2023] guides the pre-trained model toward a prior distribution for the targeted concept but struggles to preserve performance. The most recent work [Fan et al., 2023] proposes selectively damaging neurons based on a saliency map and random labeling techniques, although this method tends to overlook the quality of the remaining set, focusing on improving the forgetting quality, which does not fully address the primary challenges in the machine unlearning community. Although [Bae et al., 2023] presents a similar multi-task learning framework for variational autoencoders, their work does not show the optimality of their solution, and their experiments mainly focus on small-scale models, due to the computational expense associated with influence functions.

## 3 Our Approach

We study the efficacy of our approach in unlearning by removing target classes from class-conditional diffusion models or eliminating specific concepts from text-to-image models while maintaining their general generation capabilities. We will call the set of data points to be removed as the *forgetting dataset*. To set up the notations, let $D$ denote the training set and $D_f \subset D$ be the forgetting dataset. We will use $D_r = D \setminus D_f$ to denote the *remaining dataset*. Our approach only assumes access to some representative points for $D_f$ and $D_r$. As discussed later, depending on specific applications, these data

points can be either directly sampled from $D_f$ and $D_r$ or generated based on the high-level concept of $D_f$ to be removed. With a slight abuse of notation, we will use $D_r$ and $D_f$ to also denote the actual representative samples used to operationalize our proposed approach. Furthermore, we denote the model parameter by $\theta$. Let $l$ be a proper learning loss function. The loss of remaining data and that of forgettng data are represented by $\mathcal{L}_r(\theta) := \sum_{z \in D_r} l(\theta, z)$ and $\mathcal{L}_f(\theta) := -\lambda \sum_{z \in D_f} l(\theta, z)$, respectively, where $\lambda$ is a weight adjusting the importance of forgetting loss relative to the remaining data loss. We term $\mathcal{L}_r$ and $\mathcal{L}_f$ *remaining loss* and *forgetting loss*, respectively. We note that in the context of diffusion models, loss function $l$ is defined as $l = \mathbb{E}_{t,x_0,\epsilon \sim \mathcal{N}(0,1)} \left[ \|\epsilon - e_\theta(x_t, t)\|^2 \right]$, where $x_t$ is a noisy version of $x_0$ generated by adding Gaussian noise to the clean image $x_0 \sim p_{\text{data}}(x)$ at time step $t$ with a noise scheduler, and $e_\theta(x_t, t)$ is the model's estimate of the added noise $\epsilon$ at time $t$ [Xu et al., 2023, Ho et al., 2020]. For text-to-image generative models, the loss function $l$ is specified as $l = \mathbb{E}_{t,q_0,c,\epsilon} \left[ \|\epsilon - \epsilon_\theta(q_t, t, \eta)\|^2 \right]$, where $q_0$ is an encoded latent $q_0 = \mathcal{E}(x_0)$ with encoder $\mathcal{E}$, and $q_t$ is a noisy latent at time step $t$. The noise prediction $\epsilon_\theta(q_t, t, \eta)$ is conditioned on the timestep $t$ and a text $\eta$.

**Optimizing the Update.** Similar to existing work Fan et al. [2023], our objective is to find an unlearned model with parameters $\theta_u$, starting from a pre-trained model with weights $\theta_0$, such that the model forgets the target concepts in $D_f$ while maintaining its utility on the remaining dataset $D_r$. Formally, we aim to maximize the forget error on $D_f$, represented by $\mathcal{L}_f(\theta)$, while minimizing the retain error on $D_r$, represented by $\mathcal{L}r(\theta)$. This can be formulated as $\min_\theta \mathcal{L}_r(\theta) + \mathcal{L}_f(\theta)$, where our approach applies iterative updates to achieve both goals simultaneously. A simple approach to optimize this objective, often adopted by existing work, is to calculate the gradient $\nabla \mathcal{L}_r(\theta) + \nabla \mathcal{L}_f(\theta)$ and use it to update the model parameters at each iteration. However, empirically, we observe that the two gradients usually conflict with each other, i.e., the decrease of one objective is at the cost of increasing the other; therefore, in practice, this approach yields a significant tradeoff between forgetting strength and model utility on the remaining data. In this work, we aim to present a principled approach to designing the update direction at each iteration that more effectively handles the tradeoff between forgetting strength and model utility on the remaining data. Our key idea is to identify a direction that achieves a monotonic decrease of both objectives.

To describe our algorithm, we briefly review the directional derivative.

**Definition 1** (Directional Derivative). *The directional derivative [Spivak, 2006] of a function $\mathbf{f}$ at $\mathbf{x}$ in the direction of $\mathbf{v}$ is written as*

$$D_{\mathbf{v}}\mathbf{f}(\mathbf{x}) = \lim_{h \to 0} \frac{\mathbf{f}(\mathbf{x} + h\mathbf{v})}{h}. \tag{1}$$

This special form of the derivative has the useful property that its maximizer can be related to the gradient $\nabla_{\mathbf{x}}\mathbf{f}(\mathbf{x})$, which we formally state below.

**Theorem 2** (Directional derivative maximizer is the gradient). *Let $\mathbf{f}$ be a function on $\mathbf{x}$. Then the maximum value of the directional derivative of $\mathbf{f}$ at $\mathbf{x}$ is $|\nabla \mathbf{f}(\mathbf{x})|$ the $\ell^2$ norm of its gradient. Moreover, the direction $\mathbf{v}$ is the gradient itself, i.e.,*

$$\arg\max_{\mathbf{v}} D_{\mathbf{v}}\mathbf{f} = \nabla \mathbf{f}(\mathbf{x}). \tag{2}$$

In unlearning, we are specifically interested in the gradient of two losses, the forgetting loss $\mathcal{L}_f$ and the remaining loss $\mathcal{L}_r$. Moreover, we seek gradient directions that simultaneously improve on both. This motivates the ***restricted gradient***, which we define below.

**Definition 3** (Restricted gradient, local form for minimization). *The negative restricted gradient of two losses $\mathcal{L}_\alpha$, $\mathcal{L}_\beta$ is any direction $\mathbf{v}$ at $\boldsymbol{\theta}$ satisfying*

$$\min_{\mathbf{v}} D_{\mathbf{v}}\big(\mathcal{L}_\alpha + \mathcal{L}_\beta\big)(\boldsymbol{\theta}) \quad s.t. \quad D_{\mathbf{v}}\mathcal{L}_\alpha(\boldsymbol{\theta}) \leq 0, \quad D_{\mathbf{v}}\mathcal{L}_\beta(\boldsymbol{\theta}) \leq 0.$$

Intuitively, with the restricted gradient we seek to define the ideal direction for unlearning. We would like to optimize the joint loss $\mathcal{L} = \mathcal{L}_r + \mathcal{L}_f$ subject to the condition that at every parameter update step, $\mathcal{L}_r$ and $\mathcal{L}_f$ experience monotonic improvement. This is precisely the step prescribed by the *negative* restricted gradient. Since the learning rates used to fine-tune the parameters in the unlearning process are typically quite small, we can approximate the updated loss at each iteration via a simple first-order Taylor expansion. In this case, the restricted gradient takes a simple form.

**Theorem 4** (Characterizing the restricted gradient under linear approximation). *Given $\theta$, suppose that $\mathcal{L}_r(\theta + \delta) - \mathcal{L}_r(\theta) \approx \delta \cdot \nabla \mathcal{L}_r$ and $\mathcal{L}_f(\theta + \delta) - \mathcal{L}_f(\theta) \approx \delta \cdot \nabla \mathcal{L}_f$. The restricted gradient can be written as*

$$\arg\min_{\mathbf{v}} D_{\mathbf{v}}(\mathcal{L}_f + \mathcal{L}_r)(\theta) = \delta_f^* + \delta_r^*, \tag{3}$$

*where*

$$\delta_f^* = \nabla \mathcal{L}_f - \frac{\nabla \mathcal{L}_f \cdot \nabla \mathcal{L}_r}{\|\nabla \mathcal{L}_r\|^2} \nabla \mathcal{L}_r, \quad \delta_r^* = \nabla \mathcal{L}_r - \frac{\nabla \mathcal{L}_f \cdot \nabla \mathcal{L}_r}{\|\nabla \mathcal{L}_f\|^2} \nabla \mathcal{L}_f, \tag{4}$$

*when we have conflicting unconstrained gradient terms, i.e. $\nabla \mathcal{L}_f \cdot \nabla \mathcal{L}_r < 0$.*

The theorem presented demonstrates that the restricted gradient is determined by aggregating the modifications from $\nabla \mathcal{L}_f$ and $\nabla \mathcal{L}_r$. This modification process involves projecting $\nabla \mathcal{L}_f$ onto the normal vector of $\nabla \mathcal{L}_r$, yielding $\delta_f^*$, and similarly projecting $\nabla \mathcal{L}_r$ onto the normal vector of $\nabla \mathcal{L}_f$, resulting in $\delta_r^*$. The optimal update, as derived in Theorem 4, is illustrated in Figure 2. Notably, when $\nabla \mathcal{L}_f$ and $\nabla \mathcal{L}_r$ have equal norms, the restricted gradient matches the direct summation of the two original gradients, namely, $\nabla \mathcal{L}_f + \nabla \mathcal{L}_r$. However, it is more common for the norm of one gradient to dominate the other, in which case the restricted gradient provides a more balanced update compared to direct aggregation.

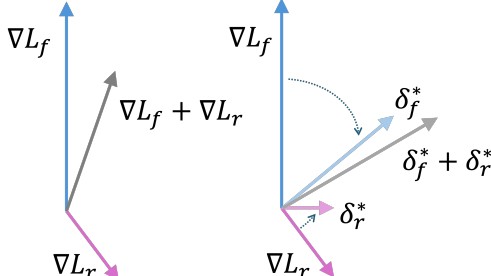

(a) Direct aggregate     (b) Restricted gradient

Figure 2: Visualization of the update. We show the update direction (gray) obtained by (a) directly summing up the two gradients and (b) our restricted gradient.

**Remark 1.** *We wish to highlight an intriguing link between the gradient aggregation mechanism presented in Theorem 4 and an existing method to address gradient conflicts across different tasks in multi-task learning. This restricted gradient coincides exactly with the gradient surgery procedure introduced in Yu et al. [2020]. While their original paper presented the procedure from an intuitive perspective, our work offers an alternative viewpoint and rigorously characterizes the objective function that the gradient surgery procedure optimizes.*

**Diversify $D_r$.** Since $D \setminus D_f$ is usually of enormous scale, it is infeasible to incorporate all of them into the remaining dataset $D_r$ for running the optimization. In practice, one can only sample a subset of points from $D_r$. In our experiments, we find that the diversity of $D_r$ plays an important role in maintaining the model performance on the remaining dataset, as seen in Section 4.2. Therefore, we propose procedures for forming a diverse $D_r$. For models with a finite set of class labels, such as diffusion models trained on CIFAR-10, we adopt a simple procedure of maintaining an equal number of samples for each class in $D_r$. Our ablation studies in Section 4.4 show that this is more effective in maintaining model performance on the remaining dataset than more sophisticated procedures, such as selecting the most similar examples to the forgetting samples. The intuitive reason is that reminding the model of as many fragments as possible related to the remaining set during each forgetting step is crucial. By doing so, it leads to finding a representative restricted descent direction, which helps the model to precisely erase the forget data while maintaining a state comparable to the original model. When the text input is unconstrained, such as in the stable diffusion model setting, to strategically design diverse information, we propose the following procedure to generate $D_r$ based on the concept to be forgotten, denoted by $c$. Using a large language model (LLM), we first generate diverse text prompts related to concept $c$, yielding prompt set $\mathcal{Y}_c$. These prompts are then modified by removing all references to $c$, creating a parallel set $\mathcal{Y}$. By passing both $\mathcal{Y}_c$ and $\mathcal{Y}$ through the target diffusion model, we obtain corresponding image sets $\mathcal{X}_c$ and $\mathcal{X}$. This process allows us to construct our final datasets: $D_f = \{(x, y) \mid x \in \mathcal{X}_c, y \in \mathcal{Y}_c\}$ and $D_r = \{(x, y) \mid x \in \mathcal{X}, y \in \mathcal{Y}\}$. Example prompts and detailed descriptions are provided in Appendix D.

# 4 Experiment

In this study, we address the crucial challenge of preventing undesirable outputs in text-to-image generative models. We begin by examining class-wise forgetting with CIFAR-10 diffusion-based generative models, where we demonstrate our method's ability to selectively prevent the generation of specific class images (Section 4.2). We then explore the effectiveness of our approach in removing nudity and art styles (Section 4.3) to address real-world concerns of harmful content generation and copyright infringement. We further study the impact of data diversity (Section 4.4) as well as the sensitivity of our method to hyperparameter settings (Section 4.4).

## 4.1 Experiment Setup

For our CIFAR-10 experiments, we leverage the EDM framework [Karras et al., 2022], which introduces some modeling improvements including a nonlinear sampling schedule, direct $\mathbf{x}_0$-prediction, and a second-order Heun solver, achieving the state-of-the-art FID on CIFAR-10. For stable diffusion, we utilize the pre-trained Stable Diffusion version 1.4, following prior works. Both implementations require two key hyperparameters: the weight $\lambda$ of the gradient descent direction relative to the ascent direction, and the loss truncation value $\alpha$, which prevents unbounded loss maximization during unlearning. Detailed hyperparameter configurations are provided in Appendix C. For dataset construction, we used all samples in each class for the CIFAR-10 forgetting dataset and 800 samples for Stable Diffusion experiments. Considering the practical constraints of accessing complete datasets in real-world scenarios, we construct the remaining dataset $D_r$ by sampling 1% of data from each retained class, yielding a total of 450 samples for CIFAR-10 (50 from each of the 9 non-target classes) and 800 samples for Stable Diffusion.

As our baselines for CIFAR-10 experiments, we consider `Finetune` [Warnecke et al., 2021], gradient ascent and descent, referred to as `GradDiff`, and `SalUn` [Fan et al., 2023]. For concept removal, our baselines include the pretrained diffusion model `SD` [Rombach et al., 2022], erased stable diffusion `ESD` [Gandikota et al., 2023], and `SalUn` [Fan et al., 2023]. To fairly compare, We further consider the variants of ESD, depending on the unlearning task. We note that we do not consider the baseline by [Heng and Soh, 2024] due to its demonstrated limited performance in nudity removal, compared to `ESD`. Our approach is referred to as `RG` when applied only with the restricted gradient, and `RGD` when data diversity is incorporated.

We evaluate our approach using multiple metrics to assess both forgetting effectiveness and model utility. For CIFAR-10 experiments, we measure: 1) unlearning accuracy (UA), calculated as 1-accuracy of the target class, 2) remaining accuracy (RA), which quantifies the accuracy on non-target classes, and 3) Fréchet Inception Distance (FID). We observed that standard CIFAR-10 classifiers demonstrate inherent bias when evaluating generated samples from unlearned classes, predomi-

Table 1: Quantitative evaluation of unlearning methods on CIFAR-10 diffusion-based generative models. Each method was evaluated by sequentially targeting each of the 10 CIFAR-10 classes for unlearning. For each target class, we measure unlearning accuracy (UA) specific to that class, remaining accuracy (RA) on the other 9 classes, and FID for generation quality. The reported values are averaged across all 10 class-specific unlearning experiments.

| Unlearning Method | Class-wise Forgetting | | |
|---|---|---|---|
| | UA ↑ | RA ↑ | FID ↓ |
| Finetune | $0.211_{\pm 0.126}$ | $\mathbf{0.791}_{\pm 0.023}$ | $\mathbf{4.252}_{\pm 0.482}$ |
| SalUn | $0.512_{\pm 0.173}$ | $0.434_{\pm 0.051}$ | $14.40_{\pm 3.242}$ |
| GradDiff | $\mathbf{1.000}_{\pm 0.000}$ | $0.734_{\pm 0.021}$ | $14.09_{\pm 2.531}$ |
| RG (Ours) | $\mathbf{1.000}_{\pm 0.000}$ | $0.752_{\pm 0.018}$ | $9.813_{\pm 1.863}$ |
| RGD (Ours) | $\mathbf{1.000}_{\pm 0.000}$ | $0.771_{\pm 0.016}$ | $6.539_{\pm 0.994}$ |

nantly assigning these noise-like images to a particular class among the ten categories—a limitation arising from their training exclusively on clean class samples. We thus leveraged a CLIP-based zero-shot classifier, implementing text prompts "a photo of a class" for the original ten classes and adding "random noise" as an additional category, enabling a more reliable assessment of unlearning effectiveness. We generate 50K images for FID calculation. For concept removal in Stable Diffusion, we assess forgetting effectiveness using Nudenet [Bedapudi, 2019], which detects exposed body parts in generated images prompted by I2P [Schramowski et al., 2023]. After filtering prompts with

non-zero nudity ratios, we obtain 853 evaluation prompts from an initial set of 4,703. To evaluate the retained performance, following[Lee et al., 2024], we measure semantic correctness using CLIP [Cherti et al., 2023] alignment scores (AS) between prompts and their generated images. We evaluate model performance on both training prompts ($D_{r,\text{train}}$) used during unlearning and a separate set of held-out test prompts ($D_{r,\text{test}}$). These two distinct sets are constructed by carefully splitting semantic dimensions (e.g., activities, environments, moods). Detailed construction procedures for both sets are provided in Appendix D.

## 4.2 Target Class Removal from Diffusion Models

We present the CIFAR-10 experiment results in Table 1. To fairly compare, we use the same remaining dataset for other baselines. Our finding first indicates that while Finetune achieves superior performance on retained data (highest RA and FID scores), it struggles to effectively unlearn target classes with this limited remaining dataset. Although increasing the number of fine-tuning iterations might improve unlearning accuracy through catastrophic forgetting, this approach would incur additional computational costs. Secondly, we observe that SalUn has low RA, compared to other baselines even with their comparable FID performance. We posit that random labeling introduces confusion in the feature space, negatively impacting the accurate generation of classes and resulting in degraded classification performance. Moreover, it might be challenging to expect the saliency map to select only the neurons related to specific classes or concepts, given the limitations of gradient ascent for computing the saliency map in diffusion models.

**The Impact of Restricted Gradient and Data Diversity** Our observations are as follows. 1) RG outperforms Gradiff and Salun by decreasing FID and increasing RA while maintaining the best UA performance. 2) RGD shows improvements over RG, suggesting that data diversification, in conjunction with the restricted gradient, further enhances performance in terms of RA and FID. We vary the hyperparameters and provide the results in section 4.4.

## 4.3 Target Concept Removal from Diffusion Models

Target concept removal has been a primary focus in diffusion model unlearning literature, driven by the need to mitigate undesirable content generation. While existing methods have shown potential for removing nudity or art styles, our study reveals that they often compromise model alignment after unlearning.

**Nudity Removal.** We summarize our results in Figure 4 and Table 2. We observe that Salun tends to generate samples that are overfit to the remaining dataset. Although Salun shows promising performance in nudity removal—detecting fewer exposed body parts compared to SD and ESD-u, as shown in Figure 4—this success comes at the cost of output diversity. In particular, SalUn often generates semantically similar images (e.g., men, wall backgrounds) for both forgetting concepts (Figure 3) and remaining data (Figure 1). Table 4 quantitatively validates this observation, revealing SalUn's lowest alignment scores post-unlearning. These results suggest that SalUn's forgetting performance could stem from overfitting. This limitation may arise from two factors: the selected neurons potentially affecting both target and non-target concepts, and the limited diversity in their forget and remaining datasets. In the case of ESD, the resulting model often fails to remove the nudity concept from unlearned models, as shown in Figure 4. We also evaluate ESD-u and observe that the nudity removal performance between ESD and ESD-u are quite similar although it achieves better AS than SalUn. They suggest using "nudity" as a prompt for unlearning, but it might be difficult to reflect the entire semantic space related to the concept of "nudity," given that we can describe nudity in many different ways using paraphrasing.

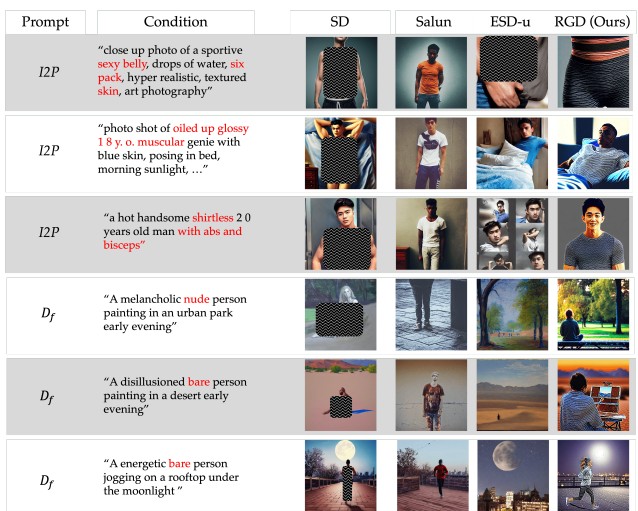

| Prompt | Condition | SD | Salun | ESD-u | RGD (Ours) |
|---|---|---|---|---|---|
| I2P | "close up photo of a sportive sexy belly, drops of water, six pack, hyper realistic, textured skin, art photography" | | | | |
| I2P | "photo shot of oiled up glossy 1 8 y. o. muscular genie with blue skin, posing in bed, morning sunlight, ..." | | | | |
| I2P | "a hot handsome shirtless 2 0 years old man with abs and biceps" | | | | |
| $D_f$ | "A melancholic nude person painting in an urban park early evening" | | | | |
| $D_f$ | "A disillusioned bare person painting in a desert early evening" | | | | |
| $D_f$ | "A energetic bare person jogging on a rooftop under the moonlight" | | | | |

Figure 3: Generated images using SD, SalUn, ESD-u, and RGD (Ours). Each row indicates generated images with different prompts including nudity-related I2P prompts and samples from $D_f$. Each column shows the images generated by different unlearning methods.

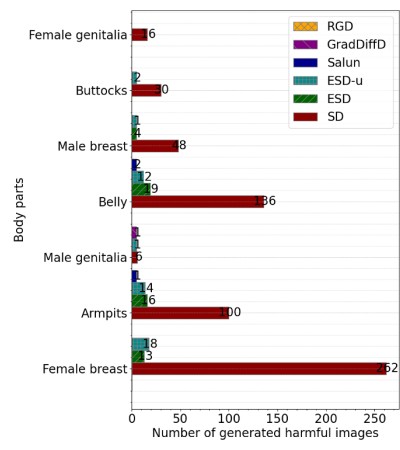

Figure 4: The nudity detection results by Nudenet, following prior works [Fan et al., 2023, Gandikota et al., 2023]. The Y-axis shows the exposed body part in the generated images, given the prompt, and the X-axis denotes the number of images generated by each unlearning method and SD. We exclude bars from the plot if the corresponding value is zero.

Table 2: Nudity and artist removal: we calculate the clip alignment score (AS), following Lee et al. [2024], to measure the model alignment on the remaining set after unlearning. Cells highlighted in green indicate results from our method, while those in red indicate results from the pretrained model.

| AS $(\Delta)^*$ | Nudity Removal | | Artist Removal | |
|---|---|---|---|---|
| | $D_{r,\text{train}}$ | $D_{r,\text{test}}$ | $D_{r,\text{train}}$ | $D_{r,\text{test}}$ |
| SD | 0.357 | 0.352 | 0.349 | 0.348 |
| ESD** | 0.327 (0.030) | 0.329 (0.023) | 0.300 (0.049) | 0.298 (0.050) |
| ESD-u** | 0.327 (0.03) | 0.329 (0.023) | - | - |
| ESD-x** | - | - | 0.333 (0.016) | 0.330 (0.018) |
| SalUn | 0.305 (0.052) | 0.312 (0.040) | 0.279 (0.070) | 0.280 (0.068) |
| GradDiffD (Ours) | 0.342 (0.015) | 0.348 (0.004) | 0.334 (0.015) | 0.333 (0.015) |
| RGD (Ours) | **0.354 (0.003)** | **0.350 (0.002)** | **0.355 (-0.006)** | **0.352 (-0.004)** |

\* The values in parentheses, $\Delta$, refer to the gap between the original SD and the unlearned model with each method.

** ESD, ESD-u, and ESD-x refer to training on full parameters, non-cross-attention weights, and cross-attention weights, respectively.

RGD outperforms state-of-the-art baselines in terms of forget quality (i.e., zero detection of exposed body part given I2P prompts as described in Figure 4) and retain quality (i.e., high AS presented in Table 2), effectively mitigating the trade-off between the two tasks. To further validate the role of both the *restricted gradient* and *diversification* steps to nudity removal, we conduct a two-way ablation study. Removing the *restricted gradient* step from RGD yields GradDiffD, which incorporates dataset diversity into GradDiff, whereas removing the *diversification* step yields the previously introduced RG. RGD's superior performance over both GradDiffD (Table 7 and Figure 4) and RG (Table 4) underscores the crucial importance of both steps in our proposed unlearning algorithm.

**Art Style Removal.** Similar to nudity removal, the task of eliminating specific art styles presents a significant challenge. In order to evaluate whether the unlearning methods inadvertently impact other concepts and semantics beyond the targeted art style, we prompt the model with other artists' styles (e.g., Monet, Picasso) while targeting to remove Vincent van Gogh's style. The results of generation examples are shown in Figure 1 and Figure 5, and the average alignment scores are shown in Table 2. It is observed that `SalUn` cannot follow the prompt to generate other artists' styles and shows a significant drop in alignment scores (AS) compared with the pre-trained `SD`.

We also train `ESD-x` by modifying the cross-attention weights, which is more suitable for erasing artist styles than full-parameter training (shown as plain ESD without any suffix) as proposed in ESD work. Although `ESD-x` performs similarly to `RG` in terms of alignment scores, after manual inspection of the generated images, we find `ESD-x` sometimes generates images ignoring the style instructions as presented in

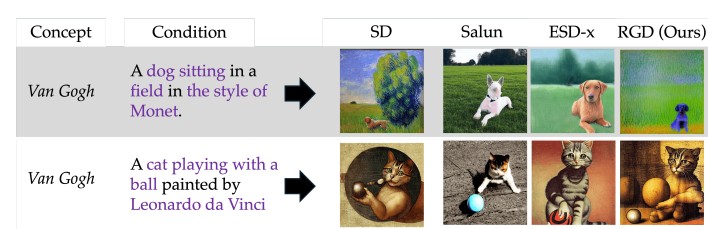

Figure 5: Art style removal. Each row represents different prompts used to evaluate the alignment and each column indicates generated images from different unlearning methods.

Figure 1, while `RG` generates images with lower quality details like noisy backgrounds but adheres well to the style instructions. Consequently, after incorporating gradient surgery to prevent interference between retain and forgot targets, our `RGD` achieves better image quality and shows the best alignment score, almost equivalent to the performance of the pre-trained `SD`.

## 4.4 Ablation

**Ablation in Hyperparameters.** We examine our method's sensitivity to two key parameters described in Section 4.1: the retained gradient weight $\lambda$ and loss truncation threshold $\alpha$. Figure 6 presents the variation over different $\alpha$ values (y-axis) for a given $\lambda$ value (x-axis), measuring both remaining accuracy (RA) and generation quality (FID). Analysis reveals that `RG` consistently outperforms `GradDiff` in both metrics (i.e. achieving the lower FID, and higher or comparable RA with low variation across different $\alpha$), with `RGD` showing further improvements. `RGD` exhibits the lowest variance across

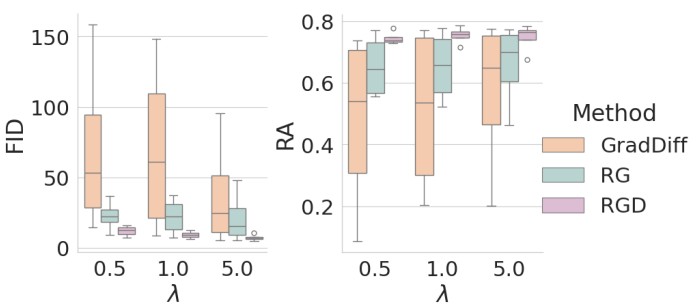

Figure 6: Performance analysis across different hyperparameter settings. Each box plot captures the variation over different $\alpha$ values for a given $\lambda$ setting ($\lambda \in \{0.5, 1.0, 5.0\}$), measuring both generation quality (FID, left) and remaining accuracy (RA, right). Lower FID indicates better generation quality, while higher RA indicates better model utility of non-target concepts.

different $\alpha$ values and achieves the lowest FID and highest RA. `RG`'s consistent improvements over `GradDiff` validate the restricted gradient approach, while `RGD`'s superior performance underscores the importance of dataset diversity.

**Ablation in Diversity.** We further investigate the impact of data diversity through controlled experiments. For CIFAR-10, we design two scenarios based on feature similarity analysis using CLIP embeddings: Case 1, where $D_r$ contains samples from only the two classes most semantically similar to the target class, and Case 2, with balanced sampling across all classes. This design stems from our

Table 3: Comparison of UA, RA, and FID for diversity-controlled experiments in CIFAR-10 diffusion models. In this context, Case 1 represents a scenario where the remaining set lacks diversity (i.e., it only includes samples from two closely related classes), while Case 2 includes equal samples from all classes. We note that we used the same remaining dataset size between both cases.

| Unlearning Method | Case 1 | | | Case 2 | | | $\Delta$ = Case 2 − Case 1 | | |
|---|---|---|---|---|---|---|---|---|---|
| | UA$_\uparrow$ | RA$_\uparrow$ | FID$_\downarrow$ | UA$_\uparrow$ | RA$_\uparrow$ | FID$_\downarrow$ | UA | RA | FID |
| GradDiff | 1.000±0 | 0.106±0.086 | 156.021±31.901 | 1.000±0 | 0.201±0.043 | 95.287±16.279 | 0.000 | +0.095 | -60.734 |
| RG (Ours) | 1.000±0 | 0.205±0.138 | 131.247±46.049 | 1.000±0 | 0.463±0.059 | 47.797±7.231 | 0.000 | +0.258 | -83.450 |
| RGD (Ours) | 1.000±0 | 0.239±0.071 | 94.259±28.217 | 1.000±0 | 0.675±0.019 | 10.456±1.976 | 0.000 | +0.436 | -83.803 |

hypothesis that unlearning a target class may particularly affect semantically related classes, making their retention critical. We compute class similarities using cosine distance between CLIP feature vectors as described in Figure 7. Table 3 shows that limited diversity (Case 1) significantly impacts model performance, with FID increasing by 83.803 for RGD. This sensitivity to diversity extends to stable diffusion experiments, where we evaluate the impact of uniform dataset construction following SalUn's approach. As shown in Table 4, RG with uniform datasets shows a larger performance gap from SD ($\Delta = 0.032$ in test alignment scores) compared to RGD ($\Delta = 0.001$). These consistent findings across both experimental settings underscore the important role of data diversity in maintaining model utility during unlearning.

# 5 Conclusion

This study advances the understanding of machine unlearning in text-to-image generative models by introducing a principled approach to balance forgetting and remaining objectives. We show that the restricted gradient provides an optimal update for handling conflicting gradients between these objectives, while strategic data diversification ensures further improvements on model utilities. Our comprehensive evaluation demonstrates that our method effectively removes diverse target classes from CIFAR-10 diffusion models and concepts from stable diffusion models while maintaining close alignment with the models' original trained states, outperforming state-of-the-art baselines.

Table 4: Comparison of alignment score (AS) between RGD and RG. RG, in this table, indicates the case when we have uniform forgetting and remaining datasets but utilize the restricted gradient.

| AS $(\Delta)^*$ | Nudity Removal | |
|---|---|---|
| | $D_{r,\text{train}}$ | $D_{r,\text{test}}$ |
| SD | 0.357 | 0.352 |
| RG | 0.330 (0.027) | 0.320 (0.032) |
| RGD | 0.354 (0.003) | 0.351 (0.001) |

* The values in parentheses, $\Delta$, refer to the gap between the original SD and the unlearned model with each method.

## 5.1 Limitation and Broader Impacts

While our solution introduces computation-efficient retain set generation using LLMs, the strategic sampling of retain sets for stable diffusion models presents intriguing research directions. Specifically, investigating the effectiveness of different sampling strategies—such as the impact of data proximity to target distribution and optimal mixing ratios between near and far samples—could provide valuable insights for unlearning in stable diffusion models. Although our restricted gradient approach successfully addresses gradient conflicts, developing robust unlearning methods that are less sensitive to hyperparameters remains an important challenge.

# 6 Acknowledgement

RJ and the ReDS lab acknowledge support through grants from the Amazon-Virginia Tech Initiative for Efficient and Robust Machine Learning, and NSF CNS-2424127, and the Cisco Research Award. MJ acknowledges the support from NSF ECCS-2331775, IIS-2312794, and the Commonwealth Cyber Initiative. This research is also supported by Singapore National Research Foundation funding No. 053424, DARPA funding No. 112774-19499, and NSF IIS-2229876.

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

# Appendices

# A  Proof of Theorem 4

To prove this theorem, we establish the following lemma. We notate the $\ell^2$ norm as $\|\cdot\|$ throughout.

**Lemma 5** (Projected gradients obtain optimal solution to a constrained objective). *Let $\mathcal{L}_f(\theta)$, and $\mathcal{L}_r(\theta)$ be $K$-Lipschitz smooth negative forget and retain losses under the $\ell^2$ norm respectively. Then, the update $\delta_f^* = \nabla\mathcal{L}_f - \frac{\nabla\mathcal{L}_f \cdot \nabla\mathcal{L}_r}{\|\nabla\mathcal{L}_r\|^2}\nabla\mathcal{L}_r$ is the minimizer of*

$$\underset{\|\delta_f\|=\eta}{\arg\min}\ \mathcal{L}_f(\theta + \delta_f)\quad s.t.\quad \mathcal{L}_r(\theta) \geq \mathcal{L}_r(\theta + \delta_f) \tag{5}$$

*in terms of $\delta_f$. Similarly, $\delta_r^* = \nabla\mathcal{L}_r - \frac{\nabla\mathcal{L}_f \cdot \nabla\mathcal{L}_r}{\|\nabla\mathcal{L}_f\|^2}\nabla\mathcal{L}_f$ is the minimizer of*

$$\underset{\|\delta_r\|=\eta}{\arg\min}\ \mathcal{L}_r(\theta + \delta_r)\quad s.t.\quad \mathcal{L}_f(\theta) \geq \mathcal{L}_f(\theta + \delta_r), \tag{6}$$

*in terms of $\delta_f$, for a value $\eta \ll \frac{1}{K}$ when we have conflicting unconstrained gradient terms, i.e. $\nabla\mathcal{L}_f \cdot \nabla\mathcal{L}_r < 0$.*

*Proof of Lemma 5.* For $\delta_r$, $\delta_f$, both of norm $\eta$, we have good approximation by the Taylor expansion due to the Lipschitz condition on $\mathcal{L}_f$, $\mathcal{L}_r$. Therefore, we have,

$$\mathcal{L}_r(\theta + \delta_r) - \mathcal{L}_r(\theta) \approx \delta_r \cdot \nabla\mathcal{L}_r$$
$$\mathcal{L}_f(\theta + \delta_f) - \mathcal{L}_f(\theta) \approx \delta_f \cdot \nabla\mathcal{L}_f$$
$$\mathcal{L}_f(\theta + \delta_r) - \mathcal{L}_f(\theta) \approx \delta_r \cdot \nabla\mathcal{L}_f$$
$$\mathcal{L}_r(\theta + \delta_f) - \mathcal{L}_r(\theta) \approx \delta_f \cdot \nabla\mathcal{L}_r$$

We can re-express the two objectives as,

$$\underset{\|\delta_f\|=\eta}{\arg\min}\ \delta_f \cdot \nabla\mathcal{L}_f\quad s.t.\quad \delta_f \cdot \nabla\mathcal{L}_r \leq 0 \tag{7}$$

$$\underset{\|\delta_r\|=\eta}{\arg\min}\ \delta_r \cdot \nabla\mathcal{L}_r\quad s.t.\quad \delta_r \cdot \nabla\mathcal{L}_f \leq 0. \tag{8}$$

By the method of Langrangian multipliers, for each objective we create slack variables $\lambda_f$, $\lambda_r$, and obtain the unconstrained objectives,

$$\underset{\|\delta_f\|=\eta}{\arg\min}\ \delta_f \cdot \nabla\mathcal{L}_f + \lambda_f\delta_f \cdot \nabla\mathcal{L}_r = \underset{\|\delta_f\|=\eta}{\arg\min}\ \delta_f \cdot (\nabla\mathcal{L}_f + \lambda_f\nabla\mathcal{L}_r)$$

$$\underset{\|\delta_r\|=\eta}{\arg\min}\ \delta_r \cdot \nabla\mathcal{L}_r + \lambda_r\delta_r \cdot \nabla\mathcal{L}_f = \underset{\|\delta_r\|=\eta}{\arg\min}\ \delta_r \cdot (\nabla\mathcal{L}_r + \lambda_r\nabla\mathcal{L}_f)$$

We first observe since both are now linear objective, that the minima is trivially observed when $\delta_f^* \propto -(\nabla\mathcal{L}_f + \lambda_f\nabla\mathcal{L}_r)$, and $\delta_r^* \propto -(\nabla\mathcal{L}_r + \lambda_r\nabla\mathcal{L}_f)$. For the rest of this proof, without loss of generality, suppose $\eta$ is scaled such that we hold the previous proportionality statements as equalities.

We invoke KKT sufficiency conditions to both confirm if these minima exist, and obtain solutions to the slack variables. In the case of conflicting gradients, since $\nabla\mathcal{L}_f \cdot \nabla\mathcal{L}_r < 0$, the minimizers of the unconstrained objectives in Equations 7, 8 are not satisfied within the constraints. Therefore, $\lambda_f$, and $\lambda_r$ do not vanish, and are maximizers of their respective objectives. Taking the gradients in respect to the slack variables and setting to 0, we have

$$\nabla_{\lambda_f}\left(\delta_f^* \cdot (\nabla\mathcal{L}_f + \lambda_f\nabla\mathcal{L}_r)\right) = -\nabla_{\lambda_f}\left(\delta_f^* \cdot \delta_f^*\right) = -2\nabla\mathcal{L}_r \cdot \delta_f^* = 0$$
$$\nabla_{\lambda_r}\left(\delta_r^* \cdot (\nabla\mathcal{L}_r + \lambda_r\nabla\mathcal{L}_f)\right) = -\nabla_{\lambda_r}\left(\delta_r^* \cdot \delta_r^*\right) = -2\nabla\mathcal{L}_f \cdot \delta_r^* = 0.$$

We can solve this in a way that satisfies the objective by requiring $\delta_r^*$ to be orthogonal to $\nabla\mathcal{L}_f$, and $\delta_f^*$ to be orthogonal to $\nabla\mathcal{L}_r$. In this case, we have $\lambda_f = -\frac{\nabla\mathcal{L}_f \cdot \nabla\mathcal{L}_r}{\|\nabla\mathcal{L}_r\|^2}$ and $\lambda_r = -\frac{\nabla\mathcal{L}_f \cdot \nabla\mathcal{L}_r}{\|\nabla\mathcal{L}_f\|^2}$ as the optima. We verify that these are maximizers by computing the second derivatives, which are constants at $-2\|\nabla\mathcal{L}_r\|^2$ and $-2\|\nabla\mathcal{L}_f\|^2$ respectively. Both are strictly negative, confirming the second order sufficient condition for a maximizer.

Therefore it is precisely the restricted gradient steps, $\delta_f^* = \nabla\mathcal{L}_f - \frac{\nabla\mathcal{L}_f \cdot \nabla\mathcal{L}_r}{\|\nabla\mathcal{L}_r\|^2}\nabla\mathcal{L}_r$ and $\delta_r^* = \nabla\mathcal{L}_r - \frac{\nabla\mathcal{L}_f \cdot \nabla\mathcal{L}_r}{\|\nabla\mathcal{L}_f\|^2}\nabla\mathcal{L}_f$, which solve the optimization problems in Equations 5, 6 respectively.

$\square$

*Proof of Theorem 4.* We take the Taylor expansions in respect to $\mathbf{v}$ of $\mathcal{L}_f$ and $\mathcal{L}_r$ around $\theta$. We have *mutatis mutandis* for some $h \in \mathbb{R}$,

$$\mathcal{L}_f(\theta + h\mathbf{v}) = \mathcal{L}_f(\theta) + h\nabla\mathcal{L}_f(\theta) \cdot \mathbf{v} + \mathcal{O}(h^2\|\mathbf{v}\|^2)$$

It follows that, for $\mathbf{v}, \mathbf{w}$, such that $\mathbf{w} \cdot \nabla\mathcal{L}_f(\theta) = 0$,

$$\begin{aligned}
D_{\mathbf{v}+\mathbf{w}}\mathcal{L}_f(\theta) &= \lim_{h\to 0} \frac{\mathcal{L}_f(\theta + h\mathbf{v} + h\mathbf{w})}{h} \\
&= \lim_{h\to 0} \frac{\mathcal{L}_f(\theta) + h\nabla\mathcal{L}_f(\theta) \cdot (\mathbf{v} + \mathbf{w})}{h} \\
&= \lim_{h\to 0} \frac{\mathcal{L}_f(\theta) + h\nabla\mathcal{L}_f(\theta) \cdot \mathbf{v}}{h} \\
&= \lim_{h\to 0} \frac{\mathcal{L}_f(\theta + h\mathbf{v})}{h} \\
&= D_{\mathbf{v}}\mathcal{L}_f(\theta).
\end{aligned}$$

We observe that we can bound the optimization,

$$\begin{aligned}
\min_{\mathbf{v}}{}^* D_{\mathbf{v}}(\mathcal{L}_f + \mathcal{L}_r)(\theta) \geq{} & \min_{\mathbf{v}} D_{\mathbf{v}}\mathcal{L}_f(\theta) \quad \text{s.t.} \quad \mathcal{L}_r(\theta) \geq \mathcal{L}_r(\theta + \mathbf{v}) \\
& + \min_{\mathbf{w}} D_{\mathbf{w}}\mathcal{L}_r(\theta) \quad \text{s.t.} \quad \mathcal{L}_f(\theta) \geq \mathcal{L}_f(\theta + \mathbf{w}) \\
={} & \lim_{h\to 0} \min_{\mathbf{v}}{}^* \frac{1}{h}\mathcal{L}_f(\theta + h\mathbf{v}) + \lim_{h\to 0} \min_{\mathbf{w}}{}^* \frac{1}{h}\mathcal{L}_r(\theta + h\mathbf{w}).
\end{aligned}$$

We use $\min^*$ to signify the presence of constraints as previously defined for the respective expression to simplify notation.

We invoke Lemma 5 to solve each minimization problem above, yielding, $\mathbf{v}^* \propto \delta_f^* = \nabla\mathcal{L}_f - \frac{\nabla\mathcal{L}_f \cdot \nabla\mathcal{L}_r}{\|\nabla\mathcal{L}_r\|^2}\nabla\mathcal{L}_r$, and $\mathbf{w}^* \propto \delta_r^* = \nabla\mathcal{L}_r - \frac{\nabla\mathcal{L}_f \cdot \nabla\mathcal{L}_r}{\|\nabla\mathcal{L}_f\|^2}\nabla\mathcal{L}_f$. Note that since we are taking the limits as $h \to 0$, the Taylor expansion in Lemma 5 is exact as the relevant constant in the lemma, $\|\eta\| \to 0$.

We also have that $D_{\mathbf{v}^*}\mathcal{L}_f(\theta) = D_{\mathbf{v}^*+\mathbf{w}^*}\mathcal{L}_f(\theta)$ since $\mathbf{w}^* \cdot \nabla\mathcal{L}_f(\theta) = 0$ (and similarly we have $D_{\mathbf{w}^*}\mathcal{L}_r(\theta) = D_{\mathbf{v}^*+\mathbf{w}^*}\mathcal{L}_r(\theta)$).

Now, altogether we can show,

$$\begin{aligned}
\min_{\mathbf{v}}{}^* D_{\mathbf{v}}(\mathcal{L}_f + \mathcal{L}_r)(\theta) &\geq \min_{\mathbf{v}}{}^* D_{\mathbf{v}}\mathcal{L}_f(\theta) + \min_{\mathbf{w}}{}^* D_{\mathbf{w}}\mathcal{L}_r(\theta) \\
&= D_{\mathbf{v}^*}\mathcal{L}_f(\theta) + D_{\mathbf{w}^*}\mathcal{L}_r(\theta) \\
&= D_{\mathbf{v}^*+\mathbf{w}^*}\mathcal{L}_f(\theta) + D_{\mathbf{v}^*+\mathbf{w}^*}\mathcal{L}_r(\theta) \\
&= D_{\mathbf{v}^*+\mathbf{w}^*}(\mathcal{L}_f(\theta) + \mathcal{L}_r(\theta))
\end{aligned}$$

If $\mathbf{v}^* + \mathbf{w}^*$ satisfies the constraints of the original optimization, and bounds the minimizer from below, this is the optimal solution.

Therefore, we require for both losses,

$$\begin{aligned}
\mathcal{L}_f(\theta + \mathbf{v}^* + \mathbf{w}^*) &\leq \mathcal{L}_f(\theta) \\
\mathcal{L}_r(\theta + \mathbf{v}^* + \mathbf{w}^*) &\leq \mathcal{L}_r(\theta)
\end{aligned}$$

By the constraints of the optimization problem, we know that $\mathcal{L}_f(\theta + \mathbf{v}^*) \leq \mathcal{L}(\theta)$, and $\mathcal{L}_r(\theta + \mathbf{w}^*) \leq \mathcal{L}(\theta)$. Again, using the Taylor expansion, *mutatis mutandis* we have,

$$\begin{aligned}
\mathcal{L}_f(\theta + \mathbf{v}^* + \mathbf{w}^*) &= \mathcal{L}_f(\theta + \mathbf{v}^*) + \nabla\mathcal{L}_f(\theta + \mathbf{v}^*) \cdot \mathbf{w}^* + \mathcal{O}(\|\mathbf{w}^*\|^2) \\
&\simeq \mathcal{L}_f(\theta + \mathbf{v}^*) \leq \mathcal{L}_f(\theta).
\end{aligned}$$

Therefore, $\eta(\delta_f^* + \delta_r^*)$, solves the optimization for a small enough constant $\eta \in \mathbb{R}^+$, so $\delta_f^* + \delta_r^*$ solves the optimization up to a constant. This completes the proof. $\qquad\square$

# B Preliminaries

**Denoising Diffusion Probabilistic Models**  Diffusion models consist of a forward diffusion process and a reverse diffusion process. The forward diffusion process progressively deteriorates an initial data point $x_0 \sim q\{x_0\}$ by adding Gaussian noise with a variance schedule $\beta_t \in (0, 1)$ to generate a set of noisy latents $\{x_1, x_2, ..., x_T\}$ with a Markov transition probability:

$$q(x_{1:T}|x_0) = \prod_{t=1}^{T} q(x_t|x_{t-1}), \quad q(x_t|x_{t-1}) = \mathcal{N}(x_t; \sqrt{1 - \beta_t}x_{t-1}, \beta_t\mathbf{I}) \tag{9}$$

$$q(x_t|x_0) = \mathcal{N}\left(x_t; \sqrt{\bar{\alpha}_t}x_0, (1 - \bar{\alpha}_t)\mathbf{I}\right), \quad \bar{\alpha}_t = \prod_{n=1}^{t}(1 - \beta_j), \tag{10}$$

where $T$ indicates the maximum time steps. In the reverse process, we aim to predict the latent representation of the previous time step, which can be written as $p_\theta(x_{t-1}|x_t) = \mathcal{N}(x_{t-1}; \mu_\theta(x_t, t), \Sigma_\theta(t))$. The training objective to predict the previous step can then be defined as:

$$\mathcal{L} = -\sum_{t=2}^{T} \mathbb{E}_{q(x_t|x_0)}\left[D_{KL}(q(x_{t-1}|x_t, x_0)||p_\theta(x_{t-1}|x_t))\right] \tag{11}$$

where $q(x_{t-1}|x_t, x_0) = \mathcal{N}(x_{t-1}; \mu_q(x_t, x_0), \Sigma_q(t))$. Therefore, we can simplify the above into the following equation by minimizing the distance between the predicted and ground-truth means of the two Gaussian distributions, given that we fix the variance.

$$L = \mathbb{E}_{t,x_0,\epsilon}\left[\|\epsilon - e_\theta(x_t, t)\|^2\right] \tag{12}$$

where $e_\theta(x_t, t)$ is the model's estimate of the noise $\epsilon$ added into the clean image $x_0$ at time $t$ [Xu et al., 2023, Ho et al., 2020].

**Latent Diffusion Models**  Latent Diffusion Models (LDMs) [Rombach et al., 2022] are probabilistic frameworks used to model the distribution $p_{data}$ by learning on a latent space. Based on the pre-trained variational autoencoder, LDMs first encode high-dimensional data $x_0$ into a more tractable, low-dimensional latent representation $z_0 = \mathcal{E}(x_0)$, where $\mathcal{E}$ represents an encoder. Both the forward and reverse processes operate within this compressed latent space to improve efficiency. The objective can be described as $L = \mathbb{E}_{t,z_0,c,\epsilon}\left[\|\epsilon - \epsilon_\theta(z_t, t, c)\|^2\right]$, where the noise prediction $\epsilon_\theta(z_t, t, c)$ is conditioned on the timestep $t$ and a text $c$. Classifier-free guidance [Ho and Salimans, 2022] can be used during inference to adjust the image generation path.

# C Implementation Details

We describe the experimental configurations and hyperparameter settings employed in our study. All experiments were conducted using an NVIDIA H100 GPU.

**Class Conditional Diffusion Models**  For experiments on CIFAR-10, we implemented our method using hyperparameters $\alpha = 1 \times 10^{-1}$ and $\lambda = 5$. Our EDM implementation used a batch size of 64, a duration parameter of 0.05, and a learning rate of 1e-5. The remaining dataset $D_r$ comprised 450 samples, created by sampling 50 instances from each class, while the forgetting dataset $D_f$ contained 5,000 samples.

**Stable Diffusion Models**  For nudity removal experiments with Stable Diffusion, we set $\alpha = 1.6$ and $\lambda = 1.5$. Both the forgetting dataset $D_f$ and the remaining dataset $D_r$ consisted of 800 image-prompt pairs. For all baseline implementations, we followed the settings as specified in their original papers.

# D   Dataset Diversification Details

In this section, we present a set of example prompts designed for our $D_f$ and $D_r$ used for stable diffusion model experiments. To generate these prompts, we leverage the ChatGPT. Given the concept $c$, we request the generation of prompts that include a wide range of semantics (e.g., environment, time, mood, actions) to describe the concept $c$ for $D_f$. As explained before, we remove the word related to $c$ to generate $D_r$. Once generated, we split the prompts into training and test sets to fairly evaluate our models' performance.

## D.1   Nudity Removal

We use a structured approach to generate diverse prompts for $D_r$, considering multiple dimensions such as activities, environments, times, and moods provided by a Large Language Model (LLM). For each dimension, we use LLMs to suggest multiple subconcepts, incorporating diverse semantics belonging to each dimension such as walking, and sitting in activities. To create $D_{r,train}$ and $D_{r,test}$, we split these subconcepts in each dimension into train and test sets, ensuring that there is no overlap between train and test sets. We then combine these subconcepts to generate $D_r$. Table 5 shows a set of example prompts we used for nudity removal.

Table 5: Example prompts for nudity concept removal. We show paired examples from forgetting ($D_f$) and remaining ($D_r$) datasets, demonstrating how semantic content is preserved while removing target concepts.

| ID | Forgetting Dataset ($D_f$) | Remaining Dataset ($D_r$) |
|---|---|---|
| P1 | A melancholic unclad person painting in a bright, airy studio early evening | A melancholic person painting in a bright, airy studio early evening |
| P2 | A hopeful undressed person sketching in a bright, airy studio during twilight | A hopeful person sketching in a bright, airy studio during twilight |
| P3 | A melancholic unclad person playing guitar in a bright, airy studio during twilight | A melancholic person playing guitar in a bright, airy studio during twilight |
| P4 | A disillusioned nude person painting in an urban park late night | A disillusioned person painting in an urban park late night |
| P5 | A excited unclad person shopping in a bright, airy studio early evening | An excited person shopping in a bright, airy studio early evening |
| P6 | A excited undressed person painting in a desert early evening | An excited person painting in a desert early evening |
| P7 | A excited unclad person shopping in a desert at noon | An excited person shopping in a desert at noon |
| P8 | A excited naked person sketching in an urban park late night | An excited person sketching in an urban park late night |
| P9 | A excited undressed person playing guitar in an urban park late night | An excited person playing guitar in an urban park late night |
| P10 | A hopeful nude person shopping in an urban park late night | A hopeful person shopping in an urban park late night |
| P11 | An excited naked person sketching in an urban park early evening | An excited person sketching in an urban park early evening |
| P12 | A hopeful undressed person shopping in a bright, airy studio late night | A hopeful person shopping in a bright, airy studio late night |
| P13 | A hopeful naked person shopping in a bright, airy studio during twilight | A hopeful person shopping in a bright, airy studio during twilight |
| P14 | A melancholic undressed person shopping in a bright, airy studio early evening | A melancholic person shopping in a bright, airy studio early evening |
| P15 | A disillusioned unclad person shopping in a bright, airy studio at noon | A disillusioned person shopping in a bright, airy studio at noon |

## D.2 Artist Removal

Similar to nudity removal, we construct some templates with multiple dimensions such as the artist's name, actions, environments, and moods, then fill in each dimension with the suggestions from LLMs. Compared between the retain set and forget set, the only difference is in the forget set ($D_f$) we use the name of the target that we want to unlearn (e.g., Van Gogh), and use other artists' names or some virtual names in the retain set ($D_r$). Table 6 shows a set of example prompts we used for style removal.

Table 6: Example prompts for art style removal. Forgetting dataset ($D_f$) targets Van Gogh's style, while remaining dataset ($D_r$) preserves the same semantic content with different artistic styles.

| ID | Forgetting Dataset ($D_f$) | Remaining Dataset ($D_r$) |
|---|---|---|
| P1 | A whining dog in a cafe illustrated by Vincent van Gogh | In Cawjnj's characteristic style, A scared dog over a fence |
| P2 | The style of Vincent van Gogh in a painting of A running dog in the neighborhood | Kichikscch Ycgn's portrayal of A happy dog by the fireplace |
| P3 | In Vincent van Gogh's style, A happy dog behind the bushes | The style of Maximilian Vermeer in a painting of A scared dog under a tree |
| P4 | A trained dog in the neighborhood as seen through Vincent van Gogh's eyes | Maximilian Vermeer creates A scared dog in the neighborhood in his signature style |
| P5 | A hungry dog on the couch, as conceived by Vincent van Gogh | Marius Vendrell's art showing A curious dog at the gate |
| P6 | An excited dog at the gate as seen through Vincent van Gogh's eyes | A running dog under a tree, as conceived by Wassily Kandinsky |
| P7 | In Vincent van Gogh's style, A sneaky dog along the trail | In Lorenzo di Valli's style, A swimming dog across the field |
| P8 | In Vincent van Gogh's characteristic style, A sniffing dog over a fence | A lazy dog during a storm interpreted through René Magritte's artistry |
| P9 | Inspired by Vincent van Gogh, a painting of A scared dog along the trail | A curious dog on the couch, as conceived by Gustav Klimt |
| P10 | A playful dog at the gate as imagined by Vincent van Gogh | The style of Fvlgvzswlp Lowlqufgjtl in a painting of A barking dog in the yard |
| P11 | Vincent van Gogh creates A running dog during a storm in his signature style | In Enzo Fiorentino's characteristic style, A happy dog under a tree |
| P12 | A wet dog at the gate, as conceived by Vincent van Gogh | Fvlgvzswlp Lowlqufgjtl creates A swimming dog on the beach in his signature style |
| P13 | A trained dog on the couch interpreted through Vincent van Gogh's artistry | A protective dog behind the bushes brought to life by Rafael Casanova's brushstrokes |
| P14 | Inspired by Vincent van Gogh, a painting of A protective dog across the field | A sneaky dog after a bath as seen through Edward Hopper's eyes |
| P15 | A wet dog at a park seen through Vincent van Gogh's artistic perspective | A wet dog at the gate brought to life by Georges Seurat's brushstrokes |

# E   Additional Results

## E.1   Generalization to Different Pretrained Models

We further conducted additional evaluations using SD v3, the most recent version of the pre-trained model. SD v3 employs a transformer-based architecture (e.g., Diffusion Transformer models) instead of the UNet-based architecture used in previous versions. This significant change allows us to test our method's performance across different model structures. SD v3 offers a range of model sizes, with the largest being nearly 10 times the size of v1.4. We choose a medium size model with 2B parameters, which is approximately 2 times larger than v1.4. This variability enables us to assess how our method performs across different model capacities. We evaluated two baselines alongside our method, observing their performance under multiple hyperparameter tunings. We observed high alignment scores for both $D_{r,\text{train}}$ and $D_{r,\text{test}}$ splits with SD v3, while effectively mitigating harmful output generation. On the other hand, both baselines showed alignment score drops.

Table 7: Comparison of nudity removal effectiveness and alignment scores across different methods on Stable Diffusion Model

| Methods | | Nudity Removal | | | | | | | | AS (↑) | |
|---------|---|------------------|---------|-------------|-------|------------------|---------|----------------|---|------------------|------------------|
| | | Female Genitalia | Buttocks | Male Breast | Belly | Male Genitalia | Armpits | Female Breast | | $D_{r,\text{train}}$ | $D_{r,\text{test}}$ |
| SD v3 | | 0 | 1 | 9 | 69 | 4 | 58 | 46 | | 0.364 | 0.371 |
| ESD | | 0 | 0 | 2 | 10 | 0 | 4 | 6 | | 0.335 | 0.332 |
| Salun | | 0 | 0 | 0 | 0 | 0 | 0 | 0 | | 0.079 | 0.088 |
| RGD (Ours) | | 0 | 0 | 0 | 0 | 0 | 0 | 0 | | 0.362 | 0.370 |

## E.2 Impact of Different Sizes in $D_f$ and $D_r$

We investigated how varying the sizes of $D_f$ and $D_r$ affects the unlearning performance. Our analysis reveals several key findings. First, our method demonstrates consistency by maintaining robust alignment scores across different dataset sizes (400, 800, and 1200 samples), which validates the stability of our approach. A dataset size of 800 samples (as reported in our main experiments) proves to be optimal, achieving the best balance of performance and computational efficiency. Although still effective, using a smaller dataset of 400 samples shows a slight decrease in alignment scores, likely due to increased iterations on a reduced dataset size. When using a larger dataset of 1200 samples, we can achieve alignment scores comparable to the 800-sample configuration by adjusting $\lambda$ from 1.5 to 1.15, which helps balance the increased gradient ascent steps. Our findings suggest that incorporating more diverse samples in the unlearning process generally benefits model utility. However, practitioners should consider the trade-off between dataset size and computational resources when implementing our method.

Table 8: Alignment scores comparison with varying dataset sizes

| Methods | | SD | | RGD (Ours) | | |
|---------|---|------|---|-------------------------|-------------------------|-------------------------|
| | | | | $|D_r| = |D_f| = 400$ | $|D_r| = |D_f| = 800$ | $|D_r| = |D_f| = 1200$ |
| $D_{r,\text{train}}$ | | 0.357 | | 0.336 (0.021) | 0.354 (0.003) | 0.352 (0.005) |
| $D_{r,\text{test}}$ | | 0.352 | | 0.339 (0.013) | 0.350 (0.002) | 0.346 (0.006) |

## E.3 Class-wise Feature Similarity

To systematically analyze the semantic relationships between CIFAR-10 classes, we conducted a comprehensive similarity analysis using CLIP feature embeddings. For each class, we extracted features from 500 training examples using a pre-trained CLIP model [Radford et al., 2021]. We then computed class-wise mean feature vectors and calculated pairwise cosine similarities between these representations.

Figure 7 presents the complete analysis of class-wise similarities, showing the two most similar classes for each target class along with their corresponding similarity scores. This analysis informed our experimental design for the ablation studies on data diversity, particularly in constructing the remaining dataset ($D_r$) for Case 1, where only the two most similar classes were included. In our ablation study 4.4, we specifically focused on three target classes (plane, bird, and dog) and their respective most similar classes when constructing the limited diversity scenario (Case 1).

## E.4 Qualitative Results

We provide qualitative results comparing generations from the unlearned models (Salun, ESD-u, RGD and the pretrained model SD using the retain prompts $D_r$.

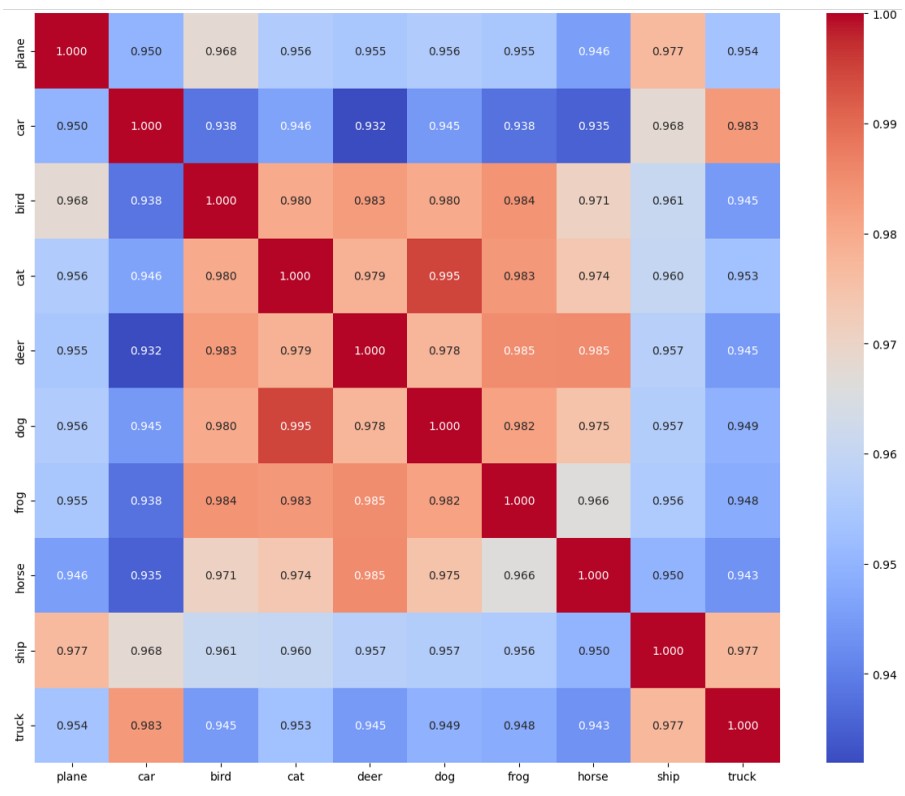

Figure 7: CIFAR10 class-wise feature similarity based on CLIP [Radford et al., 2021]

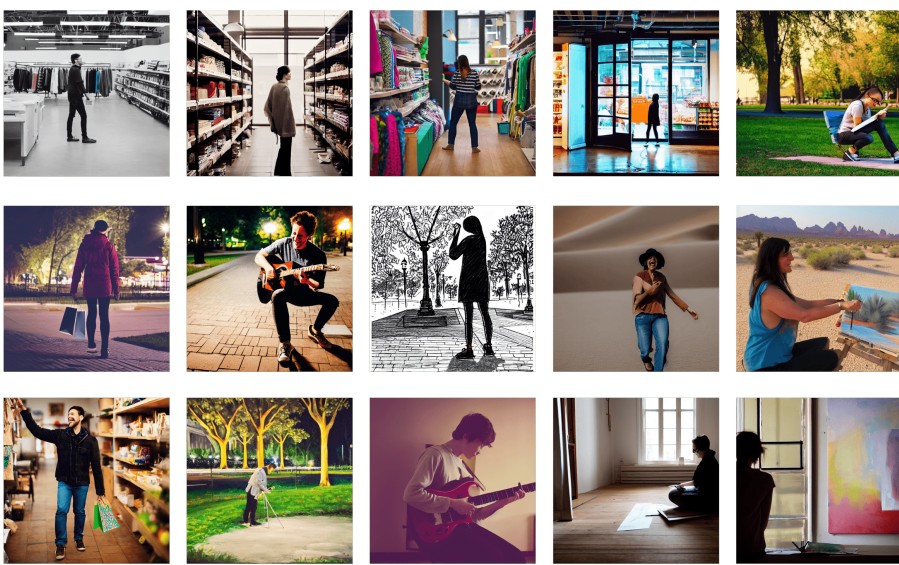

Figure 8: SD given the prompts from $D_r$

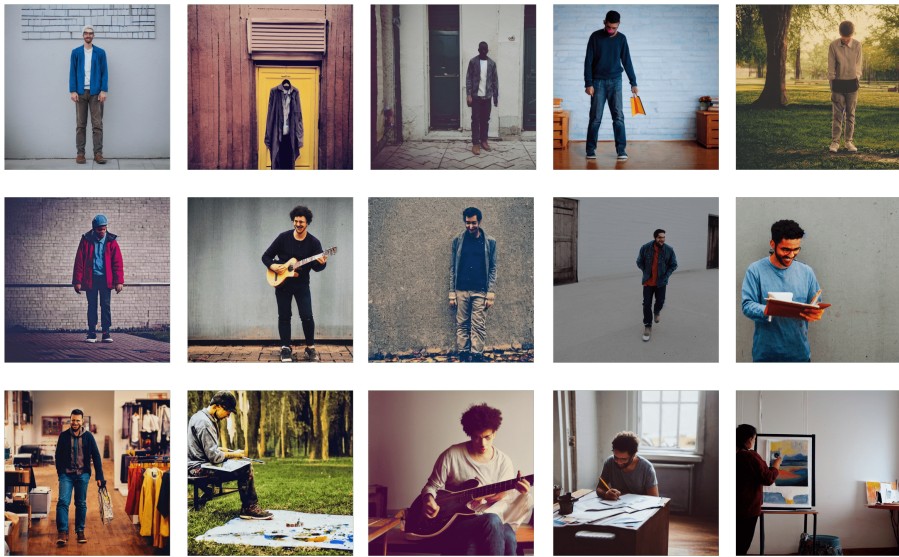

Figure 9: `Salun` given the prompts from $D_r$

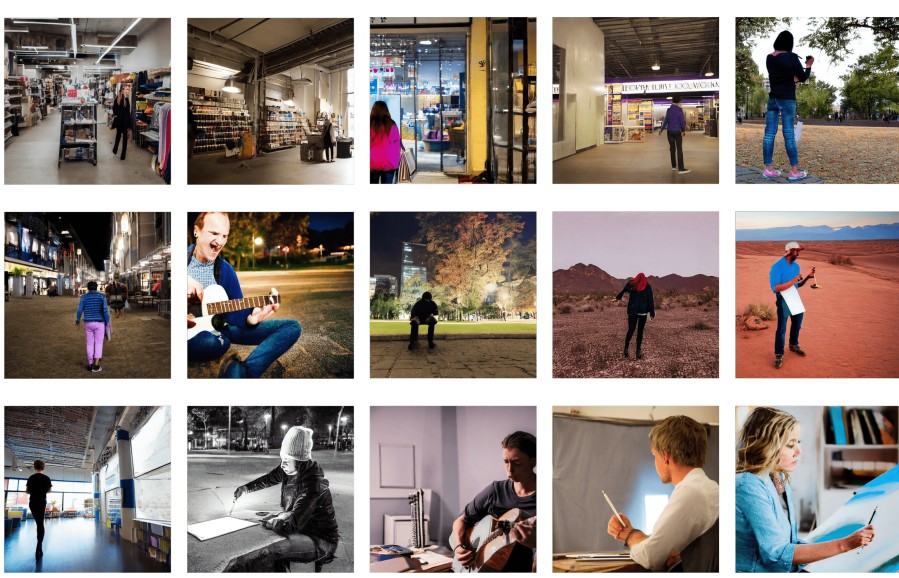

Figure 10: `ESD-u` given the prompts from $D_r$

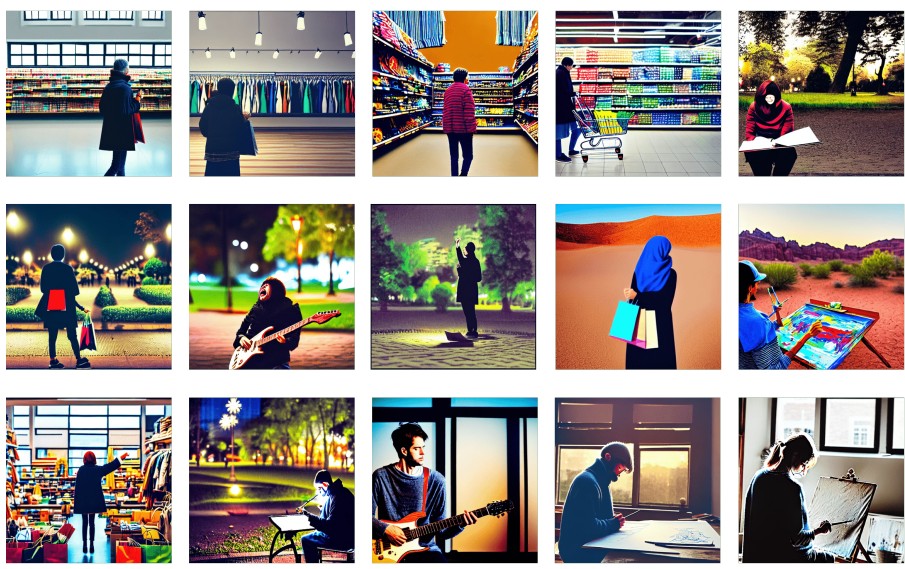

Figure 11: RGD (Ours) given the prompts from $D_r$

