# OpenReview forum: "Boosting Alignment for Post-Unlearning Text-to-Image Generative Models"
_NeurIPS.cc/2024/Conference — NeurIPS 2024 poster_

### Official Review · Reviewer_68ka · 2024-07-04

**Soundness:** 3
**Presentation:** 3
**Contribution:** 3
**Rating:** 5
**Confidence:** 3

**Summary:**

This paper presents a novel perspective on model unlearning for text-to-image generative models, considering it a constraint optimization problem. It introduces a new loss function for the unlearning process, which is a combination of the remaining loss and the forgetting loss. The paper's key technical contribution is the concept of "restricted gradient." Additionally, the proposed method uses an "LLM in the loop" approach.

**Strengths:**

- The paper offers a novel and interesting perspective on gradient surgery, viewing it as constraint optimization.
- The "LLM in the loop" approach sounds catchy.

**Weaknesses:**

- The use of the proposed loss function isn't sufficiently justified. The unlearning process employs a combination of the remaining loss and the forgetting loss, with the latter being the negative of the loss used to train the diffusion model. Making a random prediction without considering image fidelity would minimize the forgetting loss. In this work, the proposed method seems to work due to the balance between the two losses and the restricted gradient. Nonetheless, justifying the use of the forgetting loss remains challenging.
- The diversification process largely depends on the performance of LLM. For the class conditional tasks, the stratified sampling technique can work since the number of classes is quite limited. For the target concept removal tasks, it is unclear how to obtain diverse prompts that can retain the performance of the diffusion models (since there will be more than a million concepts in reality).
- This paper's primary technical contribution revolves around the concept of "restricted gradient." However, no experimental results support this technical contribution.

**Questions:**

- When training is complete, the remaining loss is zero or close to zero. Therefore, the constraints defined in definition 3 might be violated. What action does the optimizer take under such a condition?
- What is the performance of the baseline diffusion mode without any unlearning method in Table 1? Due to the classifier performance, the baseline may not achieve 100% accuracy for both UA and RA.
- How robust the clip alignment score is? Is clip alignment score well aligned with human judgement, especially in the context of artist removal?
- Do you have any detailed ablation studies on the "LLM in the loop" process? Does the number of prompts used to generate D_f and D_r affect the overall performance? If so, what would be your practical suggestions for this process?

**Limitations:**

Limitations are well addressed.

---

> ### Author Rebuttal · Authors · 2024-08-07
>
> Thank you for your insightful feedback. We have provided detailed responses to your questions and concerns below. Should you have any remaining concerns or questions, we would welcome further discussion. If our responses have adequately addressed your initial concerns, we would be grateful if you would consider adjusting your evaluation accordingly.
>
>
> **Justifying the proposed loss function. In particular, the forget loss may favor random predictions.**
>
> We appreciate your excellent point.  We would like to start by clarifying our objective and the goal of unlearning.
>
> *Problem Formulation:*
>
> Our objective is to find an unlearned model with new optimal weights $\theta_u$, given a pre-trained model with original weights $\theta$, a forget set $D_f$, and a retain set $D_r$, such that the unlearned model has forgotten $D_f$ while maintaining its utility on $D_r$. Formally, we aim to:
>
> - Maximize the forget error on $D_f$, represented by $L_f(\theta)$
> - Minimize the retain error on $D_r$, represented by $L_r(\theta)$
>
> We formulate this as: $\min_\theta L_r(\theta) + L_f(\theta)$, which is presented in line 122 in our paper.
>
> The goal of unlearning is application-dependent and should be defined and evaluated depending on applications as described in [1]. In our applications, unlearning means the unlearned model can no longer output with respect to "undesirable prompts" while preserving utility for non-target-related prompts. For example,
>
> For CIFAR10: inability to generate target class images, while still being able to generate non-target class images.
>
> For nudity or style removal: inability to generate any form of exposed body parts or target styles, while still being able to generate non-target concept-related prompts.
>
> Our choice of forget loss function for unlearning, which has also been widely used in [2,3,4], is specifically designed to maximize the loss for the forget set. This approach is justified by our goal of making the model unable to generate content related to the target concepts.
>
> **The diversification process largely depends on the performance of the LLM.**
>
>  Thank you for your insightful feedback. This is indeed an important consideration in our approach. We acknowledge that we cannot cover all concepts as a retain set, which is why it's crucial to design the retain set systematically.
>
> For target concept removal tasks, we take a different approach to address the challenge of "millions of concepts":
>
> *Generating diverse initial prompts:* We generate a retain set of prompts ($D_r$) that cover a wide range of diverse dimensions unrelated to the target concept we aim to erase. These dimensions include activities, environments, times, moods, and others suggested by a Large Language Model.
>
> *Preserving broader categories:* We aim to maintain the model's knowledge of broader categories that could potentially be affected by the unlearning process. For instance, when unlearning nudity-related concepts, we strive to preserve the model's understanding of the broader category of "a person". For example, prompts in $D_r$ might include "A person walking in a forest at sunset."
>
> To create the forget set ($D_f$), we incorporate target-related concept words into these diverse prompts. For instance, "A nude person walking in a forest at sunset".
>
> We use 'LLM in the loop' to help generate diverse prompts, but our core strategy focuses on ensuring both broader categories and diverse dimensions. This approach is based on the assumption that not all concepts are equally affected by the unlearning process. Our empirical results show that alignment scores for COCO 10K (SD: 0.334, ESD: 0.322, Δ: 0.012) have a smaller Δ than those from $D_r$ (SD: 0.352, ESD: 0.329, Δ: 0.023). This motivates our work and raises questions about the need for a detailed design of the retain set.
>
> **Experimental results do not support the primary technical contribution, the restricted gradient.**
>
> Thank you for your feedback. We would like to clarify the following points:
>
> In Table 1, GradDiff represents the method without the restricted gradient, while RG represents the method with the restricted gradient without diversification. The results demonstrate improved performance in terms of Unlearning Accuracy (UA), Remaining Accuracy (RA), and Fréchet Inception Distance (FID) when using the restricted gradient.
>
> **The restricted gradient in Definition 3 – The constraints may be violated when the loss is zero. Is this an issue?**
>
> We thank the reviewer for pointing this out. Indeed, the direction $\mathbf{v}$ will not exist when $\mathbf{x}$ is a maximizer of $L_{\alpha}$ and $L_{\beta}$.
> However, this setting does not occur in our stochastic learning setting for two reasons:
>
> 1. The losses $L_{\alpha}$ and $L_{\beta}$ are generally intractable and only approximable by stochastic minibatches. Therefore, the model parameters $\mathbf{x}$ will almost never be the maximizer on any particular minibatch, meaning that there will always be a valid direction on each training iteration.
>
> 2. During unlearning, we perform gradient ascent on the diffusion loss, which is a quadratic function with no global maximizer. Empirically, we have never reached a point where this issue has arisen.
>
> We have added a note in our manuscript to discuss these details.
>
> [1] Towards Unbounded Machine Unlearning
>
> [2] Eternal sunshine of the spotless net: Selective forgetting in deep networks
>
> [3] Unrolling sgd: Understanding factors influencing machine unlearning
>
> [4] Knowledge Unlearning for Mitigating Privacy Risks in Langauge Models

---

> ### Author Response · Authors · 2024-08-07
> **Additional Questions**
>
> We thank the reviewer for those questions. These are great comments and feedback.
>
> **Performance of the baseline diffusion model in Table 1.**
>
> Thank you for this important point. We've added the baseline SD performance on UA/RA/FID as follows:
>
> - **UA (↑):** 0.052
> - **RA (↑):** 0.955
> - **FID (↓):** 3.815
>
> We will incorporate this result in the revised version.
>
> **Robustness of the CLIP alignment score. Is it aligned with human judgment?**
>
> This is a very insightful question. We thank the reviewer for this comment. We acknowledge that there might be a gap between the clip alignment score and human judgment. Therefore, we additionally conducted the human evaluation and attached the results here.
>
> |          | Human Judgment on $D_r$ (↑) | AS on $D_r$ (↑)  |
> |----------|----------------|-----------|
> | SD       | 3.4            | 0.348     |
> | ESD      | 3.0            | 0.330     |
> | Salun    | 1.1            | 0.280     |
> | RGD (Ours) | 3.2            | 0.352     |
>
> For human judgment, we collected responses from 9 subjects who were asked to score a test set. The scoring range was from 1 (least aligned with a given prompt) to 5 (most aligned with a given prompt).
>
> Our results demonstrate that our method achieves judgment scores close to those of SD, while Salun performs poorly. The relative ranking on the retain set aligns well with the CLIP alignment score. We will incorporate these results into our revised version.
> Thank you again for your feedback.
>
> **Detailed ablation studies on the LLM in the loop process. Does the number of prompts affect performance?**
>
> We thank the reviewer for this great question. We acknowledge that it is important to study the impact of dataset size from “LLM in the loop” and therefore, we have investigated how varying the size of $D_f$ and $D_r$ affects performance. Here are our key findings:
>
> Consistency: Across different sizes (400, 800, 1200), our method maintains higher alignment scores.
>
> Optimal Size: The size of 800 (reported in our paper) shows the best balance of performance. It matches the setting used in [1], allowing for fair comparison.
>
> Smaller Set (400): While still effective, this size shows a slight decrease in alignment scores. This is likely due to simply increased iterations on a smaller dataset.
>
> Larger Set (1200): This size can achieve high alignment scores comparable to 800 if we reduce $\alpha$ from 1.5 to 1.15 to balance the increased gradient ascent steps.
>
> Therefore, the practical suggestion will be in general, it would be beneficial to include more diverse samples for unlearning to maintain the model utility.
>
> **Ablation on the size of $D_f$ and $D_r$ for Nudity Removal.**
>
> |  AS (↑)       |  $D_{r, train}$  | $D_{r, test}$  |
> |------------------------|------------------------------|------------------------------|
> | SD  | 0.357                        | 0.352                        |
> | RGD (Ours) \|$D_r$\| = \|$D_f$\| = 400 | 0.336 (0.021)               | 0.339 (0.013)               |
> | RGD (Ours) \|$D_r$\| = \|$D_f$\| = 800 | 0.354 (0.003)               | 0.350 (0.002)               |
> | RGD (Ours) \|$D_r$\| = \|$D_f$\| = 1200 | 0.352 (0.005)               | 0.346 (0.006)               |
>
>
> We believe that we have initiated the importance of proper usage of 'LLM in the loop' through this paper, and a more comprehensive study of the design will be valuable for future work.
>
> [1] SALUN: EMPOWERING MACHINE UNLEARNING VIA GRADIENT-BASED WEIGHT SALIENCY IN BOTH IMAGE CLASSIFICATION AND GENERATION

---

> > ### Comment · Reviewer_68ka · 2024-08-13
> >
> > Thanks for the additional experiments and further clarification. After reading the rebuttal and other reviews, I decided to maintain my original score.

---

### Official Review · Reviewer_DUTP · 2024-07-08

**Soundness:** 3
**Presentation:** 3
**Contribution:** 2
**Rating:** 6
**Confidence:** 4

**Summary:**

This work addresses the issue in generative models where powerful models may be generating harmful or undesired contents that should be unlearned. This work proposes to balance the unlearning objective and the text-image alignment on the remaining data, by identifying a gradient direction that achieves a monotonic decrease of both objectives (with theoretical analysis), along with a LLM-powered augmentation to encourage the diversity in the data to be unlearned. The experiments show capacity of the proposed method’s capacity in unlearning, emphasizing its retaining text-image alignment.

**Strengths:**

This paper addresses an important issue of machine unlearning, in an area when powerful generative models could produce harmful or undesired projects and should be mitigated. The idea of finding the gradient direction that is good for both objectives is sounding and interesting, although similar has been explored in other works..

Overall, this paper is easy to follow, and the experiments as well as theoretical analysis are provided in balance. The quality is good and the authors should be praised for that.

**Weaknesses:**

While this work claims to be able to do unlearning for both forgetting harmful concepts / copy-righted styles and forgetting individual objects (at least as the introduction suggests), the overall seems to be limited to the former one. For example, the diversity object is through LLM produced examples, thus  limiting to concepts that are expressible in the text, rather than obese supported by image examples. Also, the experiments do not show removing identities (e.g. celebrities), which are demonstrated in previous works.

The improvements over Salum are shown mostly in quantitative results. Qualitatively, one may argue the difference is small. This should be better explained.

**Questions:**

N/A

**Limitations:**

Yes. This work addresses the negative impact of generative models, which the work should be awarded for.

---

> ### Author Rebuttal · Authors · 2024-08-07
>
> Thank you for your insightful feedback. We have provided detailed responses to your questions and concerns below. Should you have any remaining concerns or questions, we would welcome further discussion. If our responses have adequately addressed your initial concerns, we would be grateful if you would consider adjusting your evaluation accordingly.
>
> **The authors do not provide a study on forgetting individual objects.**
>
> We appreciate the reviewer's point and concern. We would like to clarify that our experiments do not focus on forgetting individual objects, but rather on:
>
> 1. Class-wise object forgetting in diffusion models (demonstrated through CIFAR-10 experiments).
> 2. Concept removal, specifically harmful content ("nudity") and copyrighted styles ("artist style") in stable diffusion models, as described in lines 13-15.
>
> **The experiments do not show removing identities (e.g., celebrities). Do the authors have any results here?**
>
> We appreciate the reviewer's observation. We would like to emphasize that not all prior works address all applications. For example:
> - [1] focuses primarily on harmful content removal and object removal.
> - [2] addresses style and harmful content removal but not class-wise removal or celebrity removal.
>
> However, we acknowledge the importance of identity removal, particularly for celebrities, as demonstrated in previous works. Based on your valuable suggestion, we have conducted additional experiments on celebrity removal.
>
> We have also performed a human judgment evaluation to collect quantitative results on celebrity removal. For this evaluation, we gathered responses from 9 subjects who were asked to score whether the generated images contained any information regarding the target celebrity (i.e., Elon Musk). The scoring range was from 1 (Not contained) to 5 (Most contained).
>
> As shown in the following tables, we observe that our method can effectively erase the target concept, similarly to ESD. However, our CLIP alignment scores on D_r demonstrate better alignment performance after unlearning, which indicates superior performance.
>
> | AS (↑)      | $D_{r,train}$ | $D_{r,test}$ |
> |------------|-----------|----------|
> | SD         | 0.332     | 0.334    |
> | ESD        | 0.294     | 0.299    |
> | Salun      | 0.303     | 0.300    |
> | RGD (Ours) | 0.338     | 0.338    |
>
> | Human Judgment (↓) | $D_f $ |
> |-------------|-----------------------|
> | SD          | 3.3                   |
> | ESD         | 1.0                   |
> | Salun       | 2.7                   |
> | RGD (Ours)  | 1.0                   |
>
> We will incorporate these results and experiment details in our revised version. Thank you again for your great suggestion.
>
> **Qualitative improvements over SalUn appear to be small. Can the authors explain?**
>
> We appreciate your insightful feedback. We would like to clarify several points regarding our findings:
> - **Figure 3 Observations:** Salun generates visually similar images, regardless of the different forget prompts used.
> - **Figure 1 Observations:** Salun causes the unlearned model to forget many important semantic concepts, such as "quiet beach" and "sunrise". Additional qualitative examples can be found in Figure 8 in the appendix.
> - **Figure 5 Observations on Style Removal:** In style removal tasks, Salun loses closely related concepts such as Monet's style and da Vinci's style.
>
> These observations suggest that Salun's performance in reducing NSFW risk may be attributed to "overfitting" due to the uniformly designed forget and retain sets, which motivates our study on the importance of diversification in the unlearning process.
> Additionally, our experiments using the Salun method on SD v3 show further decreased alignment scores (Please find results in the general response).
> To further substantiate the qualitative comparison, we conducted a human evaluation. The results are attached below:
>
> |          | Human Judgment on $D_r$ (↑) | AS on $D_r$ (↑)  |
> |----------|----------------|-----------|
> | SD       | 3.4            | 0.348     |
> | ESD      | 3.0            | 0.330     |
> | Salun    | 1.1            | 0.280     |
> | RGD (Ours) | 3.2            | 0.352     |
>
> For human judgment, we collected responses from 9 subjects who were asked to score a test set. The scoring range was from 1 (least aligned with a given prompt) to 5 (most aligned with a given prompt). Our results demonstrate that our method achieves judgment scores close to those of SD, while Salun performs poorly. We will incorporate details about setting in our revised version. Thank you for your insightful question.
>
> [1] SALUN: Empowering Machine Unlearning via Gradient-Based Weight Saliency in Both Image Classification and Generation
> [2] Erasing Concepts from Diffusion Models

---

> > ### Comment · Reviewer_DUTP · 2024-08-13
> >
> > The authors rebuttals addressed some of my concerns. Based on that and the conversation between the authors and other reviewers, I updated my rating.

---

### Official Review · Reviewer_RWKV · 2024-07-12

**Soundness:** 2
**Presentation:** 3
**Contribution:** 2
**Rating:** 6
**Confidence:** 4

**Summary:**

This paper addresses the problem of unlearning in generative text-to-image models. They formulate a training objective that improves upon the commonly-used one that simultaneously minimizes the loss on the retain set while maximizing it on the forget set (referred to as GradDiff here); i.e. gradient descent on the retain set and ascent on the forget set, simultaneously. Specifically, as is well-established in the literature, there are trade-offs inherent in this optimization problem, e.g. maximizing the loss on the forget set may lead to “accidentally” destroying information that is permissible, with unwanted consequences like overly compromising on utility. The authors in particular emphasize a different unwanted consequence of not handling this trade-off well, that is specific to the context of text-to-image diffusion models: the generated images are no longer as well aligned with the text prompt after unlearning, compared to before. They measure this via an alignment score based on CLIP. They show that SalUn, for instance (a state-of-the-art method) actually has the lowest alignment score after unlearning, compared to other baselines.

Motivated by these issues, the authors propose a modified objective that builds on GradDiff but leads to a solution that is more balanced between the two objectives. Instead of summing the retain set gradient and the (negated) forget set gradient directly, they sum two quantities which are the projection of the former gradient on top of the normalized gradient of the latter, and the other way around. This approach (which they refer to as the “restricted gradient”) is closely related, as they discuss, to tricks used in the multi-task learning literature to jointly optimize competing objectives.

In addition, they discuss issues relating to the selection of the subset of the retain set that is used when operationalizing the training objective. They find that explicitly encouraging diversity there is important for avoiding issues relating to “overfitting” and maintaining the ability to generate diverse images that align well with textual prompts.

They show results on CIFAR-100 and using stable diffusion, comparing against different unlearning baselines, for removing a class in CIFAR-10 or a concept (e.g. nudity) from SD models. They report qualitative results (showing generations for different prompts) as well as some quantitative results, like the accuracy of the unlearned model on different sets (based on using pretrained classification models, when given as input images generated by the unlearned model), a perceptual metric for the quality of generated images as well as the alignment score. They show that their method outperforms previous baselines in terms of unlearning effectiveness (according to the particular metric) with the smallest sacrifice to alignment score and utility compared to those baselines.

**Strengths:**

- The paper tackles an important problem and uncovers previously-unknown failure modes of existing algorithms (e.g. the lack of diversity of generations coming from SalUn, indicating potential overfitting to the retain set), the drop in alignment scores after unlearning.

- The proposed objective is grounded in the multi-task literature and seems appropriate for handling competing objectives for unlearning too.

- The empirical investigation is thorough from the perspective of datasets, metrics for utility, and the authors also conduct ablations and sensitivity analyses for their method.

- The paper is for the most part well-written and easy to follow (see below for some exceptions).

**Weaknesses:**

- The paper is missing a clear problem formulation. Section 3 defines notation, but the objective isn’t stated. What is the goal of unlearning? What does it mean to unlearn something? The authors say this can be used for privacy or harmful generations or copyright, but it’s not clear how these all connect under a unified formulation and how the proposed method (and experimental design) actually addresses these problems. There is no formal definition and one must rely on the empirical metrics presented to infer how success is measured. Other types of unlearning (e.g. for privacy) are defined formally in the literature, e.g. [1,2]. Why is this definition not relevant here, since privacy is listed as one of the applications? See the below point too.

- The authors don’t present any comparisons with retrain-from-scratch which is assumed to be the gold standard for unlearning, at least according to some definitions, as discussed above (and thus serves as a reference point for the desired level of performance, e.g. how high the accuracy on the forget set should be). Several metrics have been developed, like membership inference attacks and other statistical tests according to this definition [3,4]. The authors also state that retrain-from-scratch is the gold standard solution (and that privacy is one of the applications of interest) but don’t use these metrics. It would be great to either expand the experiments accordingly or adjust the scope of claims to include only the unlearning formulation (and applications) that best corresponds to the experiments conducted here.

- Related work and baselines: [5] seems closely related to this work. It would be great to discuss the differences and compare empirically. It would also be great to compare against Safe Latent Diffusion [6]. In the experiments, how were the baselines chosen? Why not include also (Zhang et al, 2023), that was mentioned in section 2.2?

Some claims are not well substantiated:
- “these models, trained on vast amounts of public data, inevitably face concerns related to privacy” – if the training data is public, what are the privacy considerations?

- “In this study, we aim to address the crucial concerns of harmful content generation and copyright infringement [...] by focusing on the removal of target classes or concepts” – it is not clear to me how copyright infringement can be addressed by removing classes or concepts. Can you please elaborate on the setup that you have in mind?

Clarity issues relating to the experimental setup:
- For UA, why is it that higher is better? I would have thought the opposite: unlearning a concept results in poorer accuracy on classifying images of that concept. This concern is also tied to the fact that the paper is missing a problem formulation, making it hard to understand how success is defined.

- What is the rationale of evaluating model utility on the retain set only, and not the test set (e.g. in Table 1)?

- the authors discuss the size of the retain set and how it’s sampled but not the size / sampling approach of the forget set. In general, several experimental details are lacking about the benchmarks used (if they are in the appendix, please refer to the relevant sections from the main paper). In CIFAR-10, for instance, is the forget set comprised of all and only the examples of a particular class? If so, which one? In the SD experiments, how is the forget set generated?

- Table 1 caption “The metrics are averaged across all 10 classes” - the setup here is unclear; is there a different class that is unlearned each time, so UA is computed according to that one class, and then this process is repeated for unlearning different classes, and all UA’s are finally averaged?

- In Table 2, how are D_{r,train} / D_{r, test} generated for the different applications?

- “prompted by I2P” – what is I2P? Please add a citation if possible.

Minor
- “competitive nature” → “competing nature”?

- “generative reply” → “generative replay”?

- The sentence at the end of line 227 is incomplete “To fairly compare,”.


References

[1] Making AI Forget You: Data Deletion in Machine Learning. Ginart et al. 2019.

[2] Descent-to-delete: Gradient-based Methods for Machine Unlearning. Neel et al. 2020.

[3] Inexact Unlearning Needs More Careful Evaluations to Avoid a False Sense of Privacy. Hayes et al. 2024.

[4] Are we making progress in unlearning? Findings from the first NeurIPS unlearning competition. Triantafillou et al. 2024.

[5] One-dimensional adapter to rule them all: concepts, diffusion models and erasing applications. Lyu et al. 2023.

[6] Safe Latent Diffusion: Mitigating Inappropriate Degeneration in Diffusion Models. Schramowski et al. 2023.

**Questions:**

- I don’t understand how the proposed approach for diversifying the retain set in the case where the textual input is unconstrained actually achieves diversification (or to what extent). If I understand correctly, some prompts are first generated that relate to the target concept to erase, and then, to generate the retain set, these text prompts are modified for the purpose of removing words related to the target concept, and then retain images are generated from these modified prompts. But given that the prompts were initially generated to relate to the target concept, even with erasing some words, perhaps these prompts maintain some relationship to the target concept; or at least don’t uniformly cover the space of possible prompts. It seems that generating an initial set of prompts that aren’t necessarily related to the target concept (instead, they can relate to a wide range of diverse concepts) would yield more diversity. Did the authors consider this? In what ways do the authors view this approach as diversification? How does it relate to the approach usually taken in the literature for constructing the retain set?

- How different do the authors think the accuracy results would be if using a different pretrained model to assess this? Would the overall conclusions or relative rankings between methods be the same?

- The authors claim that their solution offers a monotonic improvement of each task. Is there any empirical evidence to support this claim?

Regarding my overall score / assessment, I am leaning towards weak reject due to issues with the problem description, the motivation and associated applications, and describing / justifying design decisions in the experimental protocol, tying in experiments a little better with related work (see the weaknesses section). I look forward to the author's rebuttal especially on those points.

**Limitations:**

I encourage the authors to think about potential societal impact of their work.

---

> ### Author Rebuttal · Authors · 2024-08-07
>
> Thank you for your insightful feedback. We have provided responses below and hope they clarify the points you raised. Should you have any remaining concerns or questions, we would welcome further discussion. If our responses have adequately addressed your initial concerns, we would be grateful if you would consider adjusting your evaluation accordingly.
>
> **Can the authors provide a formal definition for unlearning?**
>
> We appreciate these insightful questions, which highlight important concerns for clarification. We acknowledge that our current presentation lacks a clear problem formulation and will address this in the revised version.
>
> *Problem Formulation:*
>
> Our objective is to find an unlearned model with new optimal weights $\theta_u$, given a pre-trained model with original weights $\theta$, a forget set $D_f$, and a retain set $D_r$, such that the unlearned model has forgotten $D_f$ while maintaining its utility on $D_r$. Formally, we aim to:
>
> - Maximize the forget error on $D_f$, represented by $L_f(\theta)$
> - Minimize the retain error on $D_r$, represented by $L_r(\theta)$
>
> We formulate this as: $\min_\theta L_r(\theta) + L_f(\theta)$, which is presented in line 122 in our paper.
>
> This allows us to simultaneously pursue both objectives: forgetting specific data while preserving the utility of the remaining data.
>
> **Meaning of unlearning:**
>
> In our context, unlearning means the model can no longer output with respect to "undesirable prompts" while preserving utility for non-target related prompts. Our definition of unlearning is intentionally application-specific, aligning with the argument presented in [1]. This application-specific framework allows us to tailor our unlearning objectives to the particular needs of each use case. For example:
> - For CIFAR10: inability to generate target class images, while still being able to generate non-target class images.
> - For nudity or style removal: inability to generate any form of exposed body parts or target styles, while still being able to generate non-target concept-related prompts.
>
> Our formulation directly addresses the concerns of harmful content generation and copyright infringement. By maximizing forget error on $D_f$, we ensure the model "forgets" how to generate harmful content or copyrighted styles. Simultaneously, by minimizing retain error on $D_r$, we maintain the model's overall utility on desired tasks.
>
> **Relevance to other formal unlearning definitions for privacy:**
>
> Other types of unlearning definitions (for privacy) often rely on model indistinguishability -- comparing models trained on full dataset vs those with specific data removed. However, recent literature [2,3] challenges the effectiveness of defining unlearning success solely through this lens. These critiques argue that indistinguishability from retrain-from-scratch models may be neither sufficient nor necessary for effective unlearning across various applications.
>
> As argued in [1], it's crucial to define 'forgetting' in an application-dependent manner. For example, a privacy application might prioritize defending against Membership Inference Attacks (MIAs), potentially at the cost of some model performance. However, in the case of removing harmful concepts like nudity, maintaining model performance is a core priority. In this case, the amount of data being forgotten becomes less critical, as long as the harmful concepts are effectively removed while maintaining overall model performance. This shift necessitates different evaluation metrics, as also discussed in [1].
>
> Therefore, our application-specific approach to unlearning addresses two key challenges:
> 1. **Impracticality:** Collecting all training data related to a concept (e.g., "nudity") is often infeasible, computationally expensive, and requires additional assumptions about data access.
> 2. **Outcome-focused:** In concept removal, success is measured by the model's inability to generate target-related output, regardless of specific parameters achieved.
>
> For these reasons, we have an application-specific definition and goal of unlearning, which necessitates the use of tailored evaluation metrics to assess its effectiveness.
>
> **Lack of “retrain-from-scratch” comparisons, which are the gold standard for unlearning.**
>
> We thank you for bringing up this excellent point and constructive suggestions. We acknowledge that our initial framing may have been overly broad. Our primary application is concept removal and safe generation, rather than privacy-focused applications. We will revise the paper to clarify this focus and remove potentially misleading references to privacy as a primary application. We note that in our application, since our unlearning definition is application-specific, outcome-oriented, and aligning with [4,5] literature; in this line of work, our chosen metrics directly measure the effectiveness of concept removal and preservation of model utility. Comparing with training from scratch is not necessary for evaluation. Moreover, training from scratch is also not feasible for the scale of the models of practical interest. For example, as described in [6], with 8 Tesla V100 GPUs, training a class-guidance diffusion model takes around 2 days and we have to repeat it 10 times. Furthermore, for stable diffusion models, it is more impractical considering the size of parameters and number of training data, which is why recent works do not compare with retrain-from-scratch for diffusion models [4,5,7].
>
> [1] Towards Unbounded Machine Unlearning
>
> [2] On the necessity of auditable algorithmic definitions for machine unlearning
>
> [3] Evaluating Inexact Unlearning Requires Revisiting Forgetting
>
> [4] Erasing Concepts from Diffusion Models
>
> [5] SALUN: EMPOWERING MACHINE UNLEARNING VIA GRADIENT-BASED WEIGHT SALIENCY IN BOTH IMAGE CLASSIFICATION AND GENERATION
>
> [6] Elucidating the Design Space of Diffusion-Based Generative Models (EDM)
>
> [7] Forget-Me-Not: Learning to Forget in Text-to-Image Diffusion Models

---

> ### Author Response · Authors · 2024-08-07
> **Comparison to other baselines**
>
> We appreciate the reviewer's thoughtful suggestions regarding baseline comparisons. Our baseline selection was guided by two main principles:
>
> **Complementarity:**
> In inexact unlearning, we can broadly categorize approaches into different categories:
> 1. **Modifying the parameters:** These methods directly alter the model's weights to remove target knowledge.
> 2. **Modifying the inference:** These approaches change the inference process without altering the original model.
>
> Our work falls into the first category, focusing on modifying the model's parameters. Safe Latent Diffusion (SLD) [1] modifies the inference process to prevent certain concepts from being generated, which falls into the second category. We consider works from the second category as complementary to our approach. Moreover, the method proposed in [2] trains only an adapter and can be applied as a plug-and-play solution to other pre-trained models, which differs from our parameter-level modifications. We believe this also falls into the second category. Therefore, we did not include these methods in our comparisons, as they could potentially be used in conjunction with our method rather than as direct alternatives. However, since [6] falls into the first category, we have added additional comparison results in the general response.
>
> **State-of-the-art performance:**
> We prioritized comparisons with methods that represent the current state-of-the-art in the field. For example, both [1] and [3] have been outperformed by more recent methods such as [4] and [5], which we included in our comparisons.
>
>
> [1] Safe Latent Diffusion: Mitigating Inappropriate Degeneration in Diffusion Models
>
> [2] One-dimensional adapter to rule them all: concepts, diffusion models, and erasing applications.
>
> [3] Forget-Me-Not: Learning to Forget in Text-to-Image Diffusion Models
>
> [4] Erasing Concepts from Diffusion Models
>
> [5] SALUN: EMPOWERING MACHINE UNLEARNING VIA GRADIENT-BASED WEIGHT SALIENCY IN BOTH IMAGE CLASSIFICATION AND GENERATION
>
> [6] Ablating Concepts in Text-to-Image Diffusion Models

---

> ### Author Response · Authors · 2024-08-07
> **Some claims are not well substantiated:**
>
> **What are the privacy considerations of training on public data?**
>
> We thank the reviewer for highlighting this point. Upon reflection, we agree that our statement about privacy concerns in the context of models trained on public data might be too broad for the scope of our paper, given that our work focuses on concept removal instead of privacy. We will revise the paper to calibrate the framing.
>
> **Elaboration on how copyright infringements can be addressed by removing concepts from a model.**
>
> Thank you for your insightful question regarding the motivation and associated application of our work. Our focus on copyright infringement is motivated by recent developments in the field of generative AI. As described in [1], companies like Stability AI and MidJourney are facing civil lawsuits for training their models on artists' work without consent, enabling their models to replicate the style and work of these artists.
>
> In response to this issue, we study "style removal" as a practical application of our unlearning method on the copyright infringement application. For example, we aim to demonstrate that our method can effectively remove a particular artist's style from a model's capabilities. This removal would prevent the model from generating images in the style of the artist whose work was used without permission, thus addressing the copyright infringement issue.
>
> [1] On Copyright Risks of Text-to-Image Diffusion Models

---

> ### Author Response · Authors · 2024-08-07
> **Clarity issues relating to the experimental setup:**
>
> **UA – how come higher is better on this metric?**
>
> We thank the reviewer for pointing out the potential confusion regarding the UA metric. We define UA as 1 - accuracy of the unlearned model on $D_f$ (as noted in line 232 of our paper). This definition aims to measure the model's inability to generate or correctly classify forgotten concepts. Higher UA values indicate better unlearning performance, as they represent a greater error rate on the forget set. In other words, a higher UA suggests that the model has effectively 'forgotten' how to generate or recognize the target concepts. We adopted this metric definition to maintain consistency with previous research [1].
>
> **Rationale of evaluating model utility on the retain set only, not the test set.**
>
> Thank you for your question about our evaluation. Our evaluation pipeline for Table 1 focuses on two main aspects: the effectiveness of forgetting the target class and the preservation of model utility for the remaining classes. We don't use a separate test set in the traditional sense but rather evaluate on generated images.
>
> *Remaining Accuracy (RA)* is used to evaluate whether the unlearned model can still generate the remaining classes correctly after erasing the target class. We ask the unlearned model to generate images belonging to the remaining classes, which effectively serve as a 'test set'. We calculate the classification performance of these generated images to ensure each class of remaining images is well preserved in the unlearned model.
>
> *Unlearning Accuracy (UA)* evaluates how effectively the model has forgotten the target class. We prompt the unlearned model to generate images of the target class and then use a pre-trained classifier to determine if these generated images still belong to the target class. We will make it clear in our revised version by including more details. Thank you again for your question.
>
> **Missing experimental details for forget set construction in CIFAR-10 and SD experiments.**
>
> Thank you for your great point. We acknowledge the lack of detail regarding the forget set in our initial presentation and will incorporate these details in our revised version. For CIFAR-10 experiments: The forget set $D_f$ comprises all images of a particular target class, which is 5000 images. For Stable Diffusion (SD) experiments: Our forget set has 800 images with the corresponding prompts. The forget prompts are generated by adding the target concept (e.g., "nudity" or specific artist styles) into the retain prompts ($D_r$).
>
> **Table 2: How are $D_\text{r, train}$, $D_\text{r, test}$ generated?**
>
> We appreciate the reviewer's questions regarding the generation of $D_{r,train}$ and $D_{r,test}$ sets.
>
> *For the nudity removal application:* a) We use a structured approach to generate diverse prompts for $D_r$, considering multiple dimensions such as activities, environments, times, and moods provided by a Large Language Model (LLM). b) For each dimension, we use LLMs to suggest multiple subconcepts, incorporating diverse semantics belonging to each dimension such as walking, and sitting in activities. c) To create $D_{r,train}$ and $D_{r,test}$, we split these subconcepts in each dimension into train and test sets, ensuring that there is no overlap between train and test sets. d) We then combine these subconcepts to generate $D_r$.
>
> *For the style removal application:* similar to nudity removal, we construct some templates with multiple dimensions such as the artist’s name, actions, environments, and moods, then fill in each dimension with the suggestions from LLMs. Compared between retain set and forget set, the only difference is in the forget set ($D_f$) we use the name of the target that we want to unlearn (e.g., Van Gogh), and use other artists’ names or some virtual names in the retain set ($D_r$). We will revise the paper to incorporate these details, providing a clearer and more comprehensive explanation of our method for generating $D_{r,train}$ and $D_{r,test}$ in each application.
>
> **Table 1: How are the metrics computed across the 10 classes?**
>
> Thank you for bringing attention to this potentially confusing point. The interpretation provided is correct and reflects our intended meaning. To clarify: the caption "The metrics are averaged across all 10 classes" indeed means that we repeat the unlearning process for each class separately, and the results are averaged to produce the final metrics presented in the table. We will revise the caption to make this clearer in the revised version.
>
> **What is I2P?**
>
> We thank you for your question for further clarification. I2P stands for "Inappropriate Image Prompts", a collection of prompts designed to evaluate the effectiveness of content moderation in text-to-image models. We have cited this work in line 237 of our paper.
>
> [1] SALUN: EMPOWERING MACHINE UNLEARNING VIA GRADIENT-BASED WEIGHT SALIENCY IN BOTH IMAGE CLASSIFICATION AND GENERATION

---

> ### Author Response · Authors · 2024-08-07
> **Additional Questions.**
>
> **Explain the proposed approach for diversifying the retain set. In particular, why not generate an initial set of prompts (that aren’t necessarily related to the target concept)?**
>
> Thank you for your insightful question about our approach to diversifying the retain set.
> We fully agree with your suggestion that generating an initial set of prompts unrelated to the target concept would yield more diversity. In fact, this is precisely the approach we take.
>
> To clarify: our initial set of retain prompts ($D_r$) is generated to cover a wide range of diverse dimensions unrelated to the target concept we aim to erase. These dimensions include activities, environments, times, moods, and others. At the same time, our retain set aims to preserve the model's understanding of a broader category (e.g., "a person") that could be potentially affected by erasing “nudity.”
>
> To create $D_f$, we include target-related concept words (e.g., "nude", "naked") in the diverse prompts generated. For example, a prompt in $D_r$ might be "A person walking in a forest at sunset", while the corresponding $D_f$ prompt would be "A nude person walking in a forest at sunset". This approach ensures no direct relationship between the target concept and the retain concepts, as the initial prompts are generated independently of the target concept.
>
> We will incorporate the details of generating a diversified retain set in our revised paper.
>
> **How does it relate to the approach usually taken in the literature for constructing the retain set?**
>
> We appreciate your question about how our approach relates to existing methods in the literature. The design of the retain set has been largely overlooked in previous studies, despite its crucial role. For example,
>
> **ESD** doesn't utilize retain set information, yet claims alignment scores comparable to the original SD based on COCO dataset evaluation. However, our evaluation using "person"-related benign prompts reveals a drop in their alignment scores (as shown in Table 2), likely due to the interconnection between erasing "nudity" and general person image generation.
>
> **Salun** uses single repeated prompts for the forget (e.g., "a photo of a nude person") and retain (e.g., "a photo of a person wearing clothes") sets, which can potentially lead to overfitting.
>
> Therefore, we believe we have initiated the discussion on the importance of carefully designing the retain set, and our more systematic approach to retain set design opens up interesting avenues for further research.
>
> **How different would accuracy results be if using a different pre-trained model? Would the overall conclusions or relative rankings between methods be the same?**
>
>  Thank you for this important question about the generalizability of our results. To address this, we've conducted additional evaluations using SD v3, the most recent version of the pre-trained model. Our approach and findings are as follows (please refer to the tables in the general response):
>
> *Model Architecture:* SD v3 employs a transformer-based architecture (e.g., Diffusion Transformer models) instead of the UNet-based architecture used in previous versions. This significant change allows us to test our method's performance across different model structures.
>
> *Model Size:* SD v3 offers a range of model sizes, with the largest being nearly 10 times the size of v1.4. We choose a medium size model with 2B parameters, which is approximately 2 times larger than v1.4. This variability enables us to assess how our method performs across different model capacities.
>
> *Evaluation Approach:* We maintained the same hyperparameter settings as in v1.4 to ensure an easy generalization capability. We evaluated two baselines alongside our method, observing their performance under multiple hyperparameter tunings.
>
> *Results:* a) Alignment Scores: We observed high alignment scores for both $D_{r,train}$ and $D_{r,test}$ splits with SD v3, effectively mitigating harmful output generation. b) Baseline Comparison: Both baselines showed significant alignment score drops with multiple hyperparameter tunings, and our method continued to outperform them.

---

> ### Author Response · Authors · 2024-08-07
> **Additional Question**
>
> **The authors claim that their solution offers a monotonic improvement of each task. Is there any empirical evidence to support this claim?**
>
> We appreciate your question regarding empirical evidence for our claim of monotonic improvement. Our claim refers to consistent improvement on both objectives—forgetting the target concept (the 'forget' task) and maintaining performance on retained concepts (the 'retain' task)—without sacrificing one for the other. Theoretically, our claim for monotonic improvement is guaranteed given access to the true loss gradients in Eq. 4, and as the step size tends towards zero. In practice, there are two considerations:
>
> 1. The step size cannot be infinitely small and must be balanced against the computational budget of the training algorithm, as smaller step sizes also result in longer training times.
>
> 2. The true expected loss (i.e., the expectation of the loss over the entire data distribution) is not tractable, and we must approximate it via the empirical batch-wise loss. This results in minor deviations at each step and a stochastic approximation of the gradients in Eq. 4.
>
> To verify our claim in the empirical setting, we perform unlearning with our approach and baseline for the class-wise forgetting experiment, please see Figure 2 in the PDF:
>
> - **Forget Losses Comparison:** The left figure shows the loss for the 'forget' task. Our method maintains a higher loss compared to the baseline. This higher loss indicates better 'forgetting' of the target concept, as we want the model to perform poorly on this task.
>
> - **Retain Losses Comparison:** The right figure shows the loss for the 'retain' task. Our method maintains a relatively stable and low loss throughout the iterations, especially when compared to the baseline, which shows a sharp increase after 300 iterations. This stability is crucial as it indicates our method consistently preserves model utility on retained concepts.
>
> We note that the retain loss doesn't show a decreasing trend because our primary goal for the retain set is to maintain utility, not necessarily to improve it. The unlearning process naturally tends to increase the retain loss as we do gradient ascent. However, our "$\min D_r$" objective counteracts this increase (i.e., decrease the retain loss), resulting in the observed stability.

---

> > ### Comment · Reviewer_RWKV · 2024-08-10
> > **response to authors**
> >
> > Dear authors,
> >
> > Thank you for the thorough responses.
> >
> > I am satisfied with the discussion of the problem definition and with the promised modifications on clarifying the problem definition and the scope of the paper (in particular, discussing the application-specific definition of "Unlearning" as defined in the rebuttal, the differences from privacy definitions and reducing the scope of the claims to exclude privacy applications).
> >
> > Thank you as well for all of the the clarifications, including regarding the copyright application, the UA metric (please do emphasize more that it's 1 - the accuracy, as I found this not to be obvious), and the experimental setup. I also appreciated the additional experiments with the different pretrained model, to test generalizability, and the explanation and results re: the "monotonic improvement" question.
> >
> > I will increase my score in light of the above. Please do incorporate these clarifications in the paper (or in a section in the appendix).
> >
> > thanks!

---

> ### Author Response · Authors · 2024-08-11
> **Official Comment by Authors**
>
> Dear Reviewer RWKV,
>
> We are happy to hear that our responses have addressed your concerns and questions. We appreciate you taking the time to read our rebuttal and adjust your evaluation accordingly. We will incorporate all the clarifications, additional experimental results, and suggested modifications discussed during the rebuttal into our revised version.
> Thank you once again for your valuable, constructive feedback and for your consideration.
>
> Best regards,
>
> Authors

---

### Official Review · Reviewer_h3BB · 2024-07-14

**Soundness:** 2
**Presentation:** 3
**Contribution:** 3
**Rating:** 5
**Confidence:** 3

**Summary:**

This paper tackles the approximate machine unlearning task of the target class and concept removal from diffusion models. This work endeavors to improve the existing literature’s output quality and text alignment after unlearning. Firstly, A concept of the restricted gradient is proposed, allowing monotonic decreases of the two losses from the objectives of unlearning quality and the remaining data alignment. Secondly, data is also deliberately processed to improve its diversity, which is beneficial to text alignment. According to the two aforementioned objectives, The two components have proved effective from the ablation studies and other comparative results.

**Strengths:**

1. The paper is well-written and easy to follow, and the implementation details are also well-documented.
2. The task is well-motivated and is of practical significance to the field of safe generation using diffusion models.
3. The idea of turning a trade-off of two learning objectives into monotonic optimization is novel and useful.
4. The ablation study is a plus, for showing the effectiveness of both design components of the method.

**Weaknesses:**

1. The presentation in Table 1 is misleading, where the baseline method Finetune has the best RA and FID results; however, the results of the proposed RGD method are highlighted in bold.
2. There are two closely relevant baselines [1, 2] that need to be compared as they tackle the same task of concept removal and safe generation and have been popular as the baseline for this line of work. Missing such a comparison will lead to doubt about the proposed method’s practical significance.

[1] Ablating Concepts in Text-to-Image Diffusion Models. ICCV2023.

[2] Safe Latent Diffusion: Mitigating Inappropriate Degeneration in Diffusion Models. CVPR2023.

**Questions:**

Please see the Weaknesses section, the missing baseline comparison is my main concern.

**Limitations:**

The authors have adequately addressed the limitations and broader impacts.

---

> ### Author Rebuttal · Authors · 2024-08-07
>
> Thank you for your insightful feedback. We have provided detailed responses to your questions and concerns below. Should you have any remaining concerns or questions, we would welcome further discussion. If our responses have adequately addressed your initial concerns, we would be grateful if you would consider adjusting your evaluation accordingly.
>
> **Presentation of RA and FID results in Table 1.**
>
> Thank you for your careful review and for pointing out this discrepancy in Table 1. You are correct that the presentation could be seen as misleading, as the Finetune method indeed shows the best results for RA and FID metrics. The bolding of RGD results was intended to highlight our proposed method's overall performance across all three metrics. While Finetune excels in RA and FID, RGD shows strong performance in UA while maintaining competitive results in RA and FID. We believe this balance represents a significant advancement, especially considering the trade-offs often encountered in unlearning tasks. However, we acknowledge that our current presentation may not clearly convey this nuanced comparison. We will revise Table 1 to more accurately represent the relative strengths of each method, by highlighting the best result in each column.
>
> **Comparison to relevant baselines [1, 2].**
>
> We appreciate the reviewer's suggestion to include additional baselines. This feedback is valuable and helps us further demonstrate the significance of our work. In response, we have conducted additional experiments to compare our method with one of the baselines mentioned in [1]. Our findings are as follows:
>
> *Comparison with [1]:*
>
> We implemented [1] following their "ablating nudity" setup. Using 200 prompts related to the "anchor concept" with nudity and nsfw as caption targets, our results show it struggles to eliminate the "nudity" concept despite achieving similar alignment scores to SD. We note that their paper does not provide results on nudity concept removal. Therefore, this application likely requires further study of prompt generation and approach usage.
>
> |I2P Prompts | Female Genitalia | Buttocks | Male Breast | Belly | Male Genitalia | Armpits | Female Breast |
> |-----------------|------------------|----------|-------------|-------|----------------|---------|---------------|
> | SD              | 16               | 30       | 48          | 136   | 6              | 100     | 262           |
> | [1]             | 15               | 13       | 20          | 116   | 3              | 80      | 231           |
> | RGD (Ours)      | 0                | 0        | 0           | 0     | 0              | 0       | 0             |
>
> *Harmful Removal*
>
> | AS (↑)             | $D_{r,train}$ | $D_{r,test}$ |
> |-----------------|---------------|--------------|
> | SD              | 0.357         | 0.352        |
> | [1]             | 0.354         | 0.347        |
> | RGD (Ours)      | 0.354         | 0.350        |
>
>
> Their style removal approach doesn't well preserve close concept alignments like Monet, as shown in Figure 1, which is also described in their limitations. Also, the
>  alignment score drops after style removal.
>
>
> *Style Removal*
>
> | AS (↑)              | $D_{r,train}$ | $D_{r,test}$ |
> |-----------------|---------------|--------------|
> | SD              | 0.349         | 0.348        |
> | [1]             | 0.340         | 0.339        |
> | RGD (Ours)      | 0.355         | 0.352        |
>
> *Regarding the baseline [2]:*
>
> We note that our baseline selection was guided by two main principles:
>
> **Complementarity:** In inexact unlearning, we can broadly categorize approaches into different categories: a) Modifying the parameters: These methods directly alter the model's weights to remove target knowledge. b) Modifying the inference: These approaches change the inference process without altering the original model. Our work falls into the first category, focusing on modifying the model's parameters, and Safe Latent Diffusion (SLD) [2] modifies the inference process to prevent certain concepts from being generated, which falls into the second category. We consider works from the second category as complementary to our approach. Therefore, we did not include these methods in our comparisons, as they could potentially be used in conjunction with our method rather than as direct alternatives.
>
> **State-of-the-art performance:** We prioritized comparisons with methods that represent the current state-of-the-art in the field. For example, [2, 3] have been outperformed by more recent methods such as [4] and [5], which we included in our comparisons.
>
> [1] Ablating Concepts in Text-to-Image Diffusion Models
> [2] Safe Latent Diffusion
> [3] Forget-Me-Not: Learning to Forget in Text-to-Image Diffusion Models
> [4] Erasing Concepts from Diffusion Models
> [5] SALUN: EMPOWERING MACHINE UNLEARNING VIA GRADIENT-BASED WEIGHT SALIENCY IN BOTH IMAGE CLASSIFICATION AND GENERATION

---

> > ### Comment · Reviewer_h3BB · 2024-08-13
> >
> > I thank the authors for their rebuttal, which has well-addressed my concerns. I keep my positive recommendation for acceptance.

---

### Author Rebuttal · Authors · 2024-08-07

**General response**

We thank the reviewers for their thoughtful feedback. We are glad the reviewers find that:

- **Our paper addresses an important problem in machine unlearning** [h3BB, RWKV, DUTP]
- **Our paper is well-written and clearly presented** [h3BB, RWKV, DUTP]
- **Our proposed method is novel and effective** [h3BB, RWKV, DUTP, 68ka]

Our paper presents a new approach to unlearning in generative models, particularly text-to-image diffusion models, with a focus on balancing the unlearning objective and maintaining model utility with a restricted gradient.

**Contributions**

We introduce a novel "restricted gradient" approach that allows for the improvement of both unlearning and retain objectives.

We propose a “diversification” method to incorporate the diversity into the retain set. This approach addresses the observed failures of existing methods in maintaining model utility, an aspect that has been overlooked in previous studies.

We demonstrate improved performance in unlearning effectiveness while better-preserving model utility and text-image alignment compared to existing baselines through class-wise removal and concept removal experiments, showing the effectiveness of our method in various unlearning applications.

**Paper improvements made in response to feedback**
In response to the reviewers' feedback, we have conducted additional experiments:

We have added one more baseline for comparison  (reviewer h3BB, reviewer RWKV)

We have included experiments with an additional pre-trained model (SDv3) to evaluate the generalization of our method (reviewer RWKV)

We have conducted experiments to evaluate the improvement of our solution, compared with baseline  (reviewer RWKV)

We have included the celebrity removal experiments, along with a human study (reviewer DUTP)

We have conducted a human judgment evaluation on style removal for comparison with CLIP alignment (reviewer  68ka)

We have performed an ablation study on the size of forget and retain sets. (reviewer  68ka)

We have added the baseline diffusion model (SD v1.4)’s UA, RA, FID metrics (reviewer  68ka)

Moreover, we will incorporate details regarding the experimental setup and provide further clarifications based on reviewers’ suggestions in the revised version.

We thank the reviewers for all their insightful comments, constructive questions, and suggestions.

**Additional baseline comparison (reviewer h3BB, reviewer RWKV)**

We implemented [1] following their "ablating nudity" setup. Using 200 prompts related to the "anchor concept" with nudity and nsfw as caption targets, our results show it struggles to eliminate the "nudity" concept despite achieving similar alignment scores to SD. We note that their paper does not provide results on nudity concept removal. Therefore, this application likely requires further study of prompt generation and approach usage.

|I2P Prompts | Female Genitalia | Buttocks | Male Breast | Belly | Male Genitalia | Armpits | Female Breast |
|-----------------|------------------|----------|-------------|-------|----------------|---------|---------------|
| SD              | 16               | 30       | 48          | 136   | 6              | 100     | 262           |
| [1]             | 15               | 13       | 20          | 116   | 3              | 80      | 231           |
| RGD (Ours)      | 0                | 0        | 0           | 0     | 0              | 0       | 0             |

*Harmful Removal*

| AS (↑)            | $D_{r,train}$ | $D_{r,test}$ |
|-----------------|---------------|--------------|
| SD              | 0.357         | 0.352        |
| [1]             | 0.354         | 0.347        |
| RGD (Ours)      | 0.354         | 0.350        |

Their style removal approach doesn't well preserve close concept alignments like Monet, as shown in Figure 1, which is also described in their limitations. Also, the $D_r$ alignment score drops after style removal.

*Style Removal*

| AS (↑)              | $D_{r,train}$ | $D_{r,test}$ |
|-----------------|---------------|--------------|
| SD              | 0.349         | 0.348        |
| [1]             | 0.340         | 0.339        |
| RGD (Ours)      | 0.355         | 0.352        |

[1] Ablating Concepts in Text-to-Image Diffusion Models

**Additional generalization evaluation - evaluate on SDv3 (reviewer RWKV)**

| I2P Prompts | Female Genitalia | Buttocks | Male Breast | Belly | Male Genitalia | Armpits | Female Breast |
|-----------------|------------------|----------|-------------|-------|----------------|---------|---------------|
| SD v3            | 0                | 1        | 9           | 69     | 4              | 58       | 46             |
| ESD             | 0                | 0        | 2           | 10    | 0              | 4       | 6             |
| Salun           | 0                | 0        | 0           | 0     | 0              | 0       | 0             |
| RGD (Ours)      | 0                | 0        | 0           | 0     | 0              | 0       | 0             |

| AS (↑) | $D_{r,train}$ | $D_{r,test}$ |
|-----------------------------------|---------------|--------------|
| SD v3                              | 0.364         | 0.371        |
| ESD                               | 0.335         | 0.332        |
| Salun                             | 0.079         | 0.088        |
| RGD (Ours)                        | 0.362         | 0.370        |

We've conducted additional evaluations using SD v3, the most recent version of the pre-trained model. Our findings are as follows: 1) We maintained the same hyperparameter settings as in v1.4 to ensure an easy generalization capability. 2) We evaluated two baselines alongside our method under multiple hyperparameter tunings.

We observed high alignment scores for both $D_{r,train}$ and $D_{r,test}$ splits with SD v3, while effectively mitigating harmful output generation. On the other hand, both baselines showed alignment score drops.

---

### Decision · Program_Chairs · 2024-09-25

**Decision:**

Accept (poster)

**Comment:**

The reviewers highlighted several key strengths of the paper, including the significance of the addressed problem, the soundness of the proposed method, easy to follow writing, thoroughness of the empirical validation, and the balanced approach between experimental results and theoretical justifications.

Reviewers also noted some issues, such as the need for additional empirical validation, improvements in presentation, further discussion of the problem definition, and more detailed justifications for certain design decisions in the experimental protocol. These concerns were adequately addressed by the authors during the rebuttal and subsequent discussions.

Furthermore, the authors  positioned well their work within the relevant literature, notably citing arXiv:2001.06782, which employed a similar gradient surgery technique in multitask learning. It might be beneficial for readers of the paper to mention also that a similar gradient orthogonalization method has recently been applied in disentanglement learning, as discussed in arXiv:2308.12696.

In light of the above, acceptance is recommended. The authors are strongly encouraged to incorporate the proposed improvements into the final version of the paper.